Workshop at the 6th Symposium on Advances in Approximate Bayesian Inference (non-archival), 2024 1–33

# Towards Model-Agnostic Posterior Approximation for Fast and Accurate Variational Autoencoders

**Yaniv Yacoby**                                                              YY109@WELLESLEY.EDU
*Wellesley College*

**Weiwei Pan**                                                              WEIWEIPAN@G.HARVARD.EDU
**Finale Doshi-Velez**                                                          FINALE@SEAS.HARVARD.EDU
*Harvard University*

**Keywords:** Variational Autoencoders, Approximate Inference, Non-Identifiability

## 1. Introduction

Variational Autoencoders (VAEs) (Kingma and Welling, 2014) are deep generative latent variable models that transform a simple distribution over a latent space into a complex data distribution. VAE inference consists of learning two components: (1) a generative model, which transforms a simple distribution over a latent space into the distribution over observed data, and (2) an inference model, which approximates the posterior of the latent codes given data. The two components are jointly learned by optimizing a lower bound to the generative model's log marginal likelihood (LML). In early phases of joint training, the inference model poorly approximates the latent code posteriors. Recently, He et al. (2019) showed that this leads optimization to get stuck in local optima, negatively impacting the learned generative model. To mitigate this issue, He et al. (2019) suggest ensuring a high-quality inference model via iterative training: maximizing the objective function relative to the inference model before every update to the generative model.Unfortunately, iterative training is inefficient, requiring heuristic criteria for reverting from iterative training back to joint training for speed. One way to speed up non-joint VAE inference is to train the generative and inference models independently, for example, by analytically computing the posterior of a given generative model. However, there is no systematic way to analytically approximate high-quality inference models for arbitrary generative models. In this work, we suggest an alternative VAE inference algorithm that trains the generative and inference models *independently*. Specifically, we propose a method to approximate the posterior of the *true* model a priori; fixing this posterior approximation, we then maximize the lower bound relative to only the generative model. We note that, by conventional wisdom, this approach should rely on the true prior and likelihood of the true model to approximate its posterior, both of which are unknown. In this work, we show that we can, in fact, compute a deterministic, model-agnostic posterior approximation (MAPA) of the *true* model's posterior. We then use MAPA to develop a proof-of-concept inference method for VAEs. We present preliminary results on low-dimensional synthetic data that (1) MAPA captures the trend of the true posterior, and (2) our MAPA-based inference method performs better density estimation with less computation than baselines. Lastly, we present a roadmap for scaling the MAPA-based inference method to high-dimensional data.

## 2. Background and Notation

We are given $N$ observations $x_n \in \mathbb{R}^D$, generated from corresponding latent codes $z_n \in \mathbb{R}^L$ with $L < D$. We define $X = \{x_n\}_{n=1}^N$ and $Z = \{z_n\}_{n=1}^N$.

**Model.**   We assume our observed data was generated as follows:

$$z \sim p_z(\cdot; \psi), \quad x|z \sim p_{x|z}(\cdot|f_\theta(z)), \tag{1}$$

where $\psi$ parameterizes the prior and $\theta$ parameterizes a NN, $f_\theta(\cdot)$. We define $\log p_x(x; \theta, \psi) = \log \mathbb{E}_{z \sim p_z(\cdot; \psi)} \left[ p_{x|z}(x|f_\theta(z)) \right]$ to be the LML. We refer to $\theta^{\text{GT}}, \psi^{\text{GT}}$ as the parameters of the ground-truth, data-generating model, and $z_n^{\text{GT}}$ as the ground-truth latent codes that generated $x_n$ (similarly, $Z^{\text{GT}} = \{z_n^{\text{GT}}\}_{n=1}^N$).

**Inference.**   Our goal is to maximize the observed data LML $p_x(X; \theta, \psi)$. Since it is intractable, we instead maximize the IWAE stochastic bound (Burda et al., 2016):

$$\log p_x(x_n; \theta, \psi) \geq \mathbb{E}_{z^{(1)}, \ldots, z^{(S)} \sim q_{z|x}(\cdot|x_n; \phi)} \underbrace{\left[ \log \frac{1}{S} \sum_{s=1}^S \frac{p_{x|z}(x_n|f_\theta(z^{(s)})) \cdot p_z(z^{(s)}; \psi)}{q_{z|x}(z^{(s)}|x_n; \phi)} \right]}_{\text{stochastic lower bound, } \mathcal{L}_{\text{IWAE}}^S(x_n; \theta, \psi, \phi)}, \tag{2}$$

where $q_{z|x}(\cdot|x_n; \phi)$ is the proposal distribution, whose parameters $\phi$ are jointly optimized with $\theta$. This bound has two notable properties. First, it monotonically tightens as the number of importance samples $S$ increases, becoming tight when $S \to \infty$ (Burda et al., 2016). Second, the tightness of the bound is proportional to the variance of the importance sampling scheme (Domke and Sheldon, 2018). For this bound, a naive implementation with auto-differentiation results in noisy gradients with respect to $\phi$, requiring a specialized gradient estimator (Roeder et al., 2017; Tucker et al., 2019).

## 3. "Empiricalized" Models

**Empiricalized model.**   Given the original generative model from Eq. (1), we "empiricalize" it, meaning we replace the prior $p_z(\cdot, \psi)$ with an *empirical* distribution:

$$z_m \sim p_z(\cdot; \psi) \text{ for } m \in [1, M], \quad i_n \sim p_i(\cdot) = \mathcal{U}[1, \ldots, M], \quad x_n|i_n, Z \sim p_{x|i, Z}(\cdot|f_\theta(z_{i_n})). \tag{3}$$

Under this new generative process, we assume we've already sampled $M$ draws from the prior $p_z(\cdot, \psi)$. We then use index $i_n$, drawn at uniform, to select which latent code to decode. $Z$ is therefore the prior's *hyperparameter*. The empiricalized model is similar in spirit to bootstrapping, converging to the original generative process as $M \to \infty$. From here on, we set that $M = N$, since our inference will leverage this for efficiency.

**Maximizing the LML.**   Whereas in the original model, the LML requires a marginalization over the latent code $z$, empiricalized models require marginalization over the latent indices $i$:

$$\log p_x(x; \theta, Z) = \log \mathbb{E}_{i \sim p_i(\cdot)} \left[ p_{x|i, Z}(x|f_\theta(z_i)) \right] = \log \frac{1}{N} \sum_{i=1}^N p_{x|i, Z}(x|f_\theta(z_i)). \tag{4}$$

Here, we can think of $Z$ as the hyper-parameter of the prior, equivalent to $\psi$ of the original prior $p_z(\cdot; \psi)$; the locations of latent codes control the shape of the prior. If we were to fix $z_n$ to a grid in the latent space, at this point in the derivation, it would resemble a Generative Topographic Mapping (Bishop et al., 1998). Our goal, however, is to maximize $\log p_x(X; \theta, Z)$ relative to $\theta$ *and* $\psi$ (which in this case, refers to $Z$):

$$\theta^*, Z^* = \mathrm{argmax}_{\theta, Z} \frac{1}{N} \sum_{n=1}^{N} \log \frac{1}{N} \sum_{i=1}^{N} p_{x|i,Z}(x_n | f_\theta(z_i)). \tag{5}$$

Given $Z^*$, we can fit a model (e.g. Normalizing Flow (Kobyzev et al., 2020)) to $Z^*$ to obtain a parametric $p_z(\cdot; \psi)$. Since for our empiricalized model to converge to the original model, $N$ needs to be sufficiently large, the inner sum in Eq. (5) becomes expensive to compute. In Section 5, we introduce a novel inference method that circumvents this issue.

**Prior amortization.** We amortize the *prior* latent codes $z_n \in Z$ with a NN, $z_n = g_\psi(x_n)$, parameterized by $\psi$:

$$\theta^*, \psi^* = \mathrm{argmax}_{\theta, \psi} \frac{1}{N} \sum_{n=1}^{N} \log \frac{1}{N} \sum_{i=1}^{N} p_{x|i,Z}(x_n | f_\theta \circ g_\psi(x_i)). \tag{6}$$

The above can be thought of as an "amortized mixture model," where the mixture components are parameterized by an autoencoder (AE), $f_\theta \circ g_\psi(\cdot)$, to lie on a lower-dimensional manifold. At a high level, this amortization resembles non-parametric priors for VAEs (e.g. Tomczak and Welling (2018)), used to match the prior to the aggregated posterior. In contrast, the amortization here serves other practical purposes. First, it increases the efficiency of training, since gradients relative to $\psi$ help shape the entire prior (whereas gradients with respect to a batch of $z_n$'s do not). Second, it provides a convenient mapping to and from the latent space, which is useful downstream. Lastly, it prevents overfitting by ensuring that the latent codes lie on a well-behaved low-dimensional manifold. In Appendix B.1, we show a new relationship between Eq. (6) and the training objective of an AE—that the AE objective is a lower bound.

## 4. Deterministic Model-Agnostic Posterior Approximation (MAPA)

So why perform approximate inference on the empiricalized model as opposed to on the original model? Because this will allow us to *estimate the probability of a latent code's index $i_n$ independently of its location in latent space $z_n$*. Now, we leverage this trick to propose a deterministic, model-agnostic posterior approximation (over indices) of the *true* empiricalized model,

$$p_{i|x,Z}(i|x; \theta^{\mathrm{GT}}, Z^{\mathrm{GT}}) = \frac{p_{x|i,Z}(x|f_{\theta^{\mathrm{GT}}}(z_i^{\mathrm{GT}})) \cdot p_i(i)}{\sum\limits_{j=1}^{N} p_{x|i,Z}(x|f_{\theta^{\mathrm{GT}}}(z_j^{\mathrm{GT}})) \cdot p_i(j)} = \frac{p_{x|i,Z}(x|f_{\theta^{\mathrm{GT}}}(z_i^{\mathrm{GT}}))}{\sum\limits_{j=1}^{N} p_{x|i,Z}(x|f_{\theta^{\mathrm{GT}}}(z_j^{\mathrm{GT}}))}. \tag{7}$$

**Insight.** Even without knowing the ground-truth decoder $f_{\theta^{\mathrm{GT}}}(\cdot)$ of the empiricalized generative model, we already know *something* about $p_{i|x,Z}(i|x; \theta^{\mathrm{GT}}, Z^{\mathrm{GT}})$. Consider two observations $x_n, x_m$ and the corresponding indices $i_n, i_m$ that generated them. If $x_n$ and $x_m$ are

"far" from each other (according to the likelihood), the posteriors $p_{i|x,Z}(i_m|x_n; \theta^{\text{GT}}, Z^{\text{GT}})$ and $p_{i|x,Z}(i_n|x_m; \theta^{\text{GT}}, Z^{\text{GT}})$ are likely to be low, while the posteriors $p_{i|x,Z}(i_n|x_n; \theta^{\text{GT}}, Z^{\text{GT}})$ and $p_{i|x,Z}(i_m|x_m; \theta^{\text{GT}}, Z^{\text{GT}})$ should be high. Fig. 1 depicts this exactly: in the figure, the three nearby observations $x_1, x_2, x_3$ all have similar posteriors, under which *all three* latent codes $z_1, z_2, z_3$ have a high score. In contrast, $x_4$, which is far from the first three observations, has a different posterior and its latent code $z_4$ have a low score under their posteriors. This confirms the intuition behind MAPA, in which the posteriors of nearby observations should score their corresponding latent code with high probability. Moreover, this behavior holds across multiple choices of decoder $f_\theta(\cdot)$, meaning it is robust to model non-identifiability. We will now incorporate this insight into an approximation of the ground-truth posterior (without knowing the ground-truth prior or decoder $f_{\theta^{\text{GT}}}(\cdot)$), using some notion of "distance" between observations.

**MAPA.** We define MAPA as a categorical distribution with the $i$th probability set to:

$$p_{i|x,Z}(i|x; \theta^{\text{GT}}, Z^{\text{GT}}) \approx \frac{\kappa(x_n|x_i)}{\sum\limits_{j=1}^{N} \kappa(x_n|x_j)} = q_{i|n}(i|n), \tag{8}$$

where $\kappa(\cdot|\cdot)$ represents a notion of proximity between observations (though it need not be symmetric). This approximation is "model-agnostic" because it does not depend on the choice of prior or likelihood, though in practice, we select $\kappa(\cdot|\cdot)$ in accordance with the observation noise distribution. Moreover, it can be *computed once per data-set and cached.* See Appendix B.2 for the derivation, which starts with the left-hand form (given all ground-truth parameters) and ends with the right-hand form (independent of the true prior and likelihood, and of the ground-truth parameters). In Section 6, we demonstrate that MAPA captures the trend of the ground-truth empiricalized and original posteriors.

We note that MAPA resembles a Kernel Density Estimator (KDE) (Chen, 2017). As such, one might wonder: how would this scale to high-dimensional data? Whereas KDEs use distance in observation-space to approximate a distribution over (high-dimensional) observation-space, MAPA uses these distances to approximate a posterior distribution over a *low-dimensional latent space.* MAPA also bears similarity to Approximate Bayesian Computation (Sisson et al., 2018) in using distances between observations generated from the prior (or original generative process) to estimate a posterior over unobserved variables.

## 5. Proof-of-Concept: MAPA-based Inference

**MAPA-based stochastic lower bound.** We now leverage $q_{i|n}(i|n)$ to derive a lower bound to the LML of the empiricalized model from Eq. (6). We define $\mathcal{B}_n(k)$ to be the set of $k \in \{1, \ldots, N\}$ indices for which $q_{i|n}(\cdot|n)$ is largest and $\tilde{q}_{i|n}^k(i|n)$ to be $q_{i|n}(\cdot|n)$ to be renormalized after setting the probability of its $k$ largest elements to 0:

$$\tilde{q}_{i|n}^k(i|n) = \frac{q_{i|n}(i|n) \cdot \mathbb{I}[i \notin \mathcal{B}_n(k)]}{\sum\limits_{j=1}^{N} q_{i|n}(j|n) \cdot \mathbb{I}[j \notin \mathcal{B}_n(k)]}. \tag{9}$$

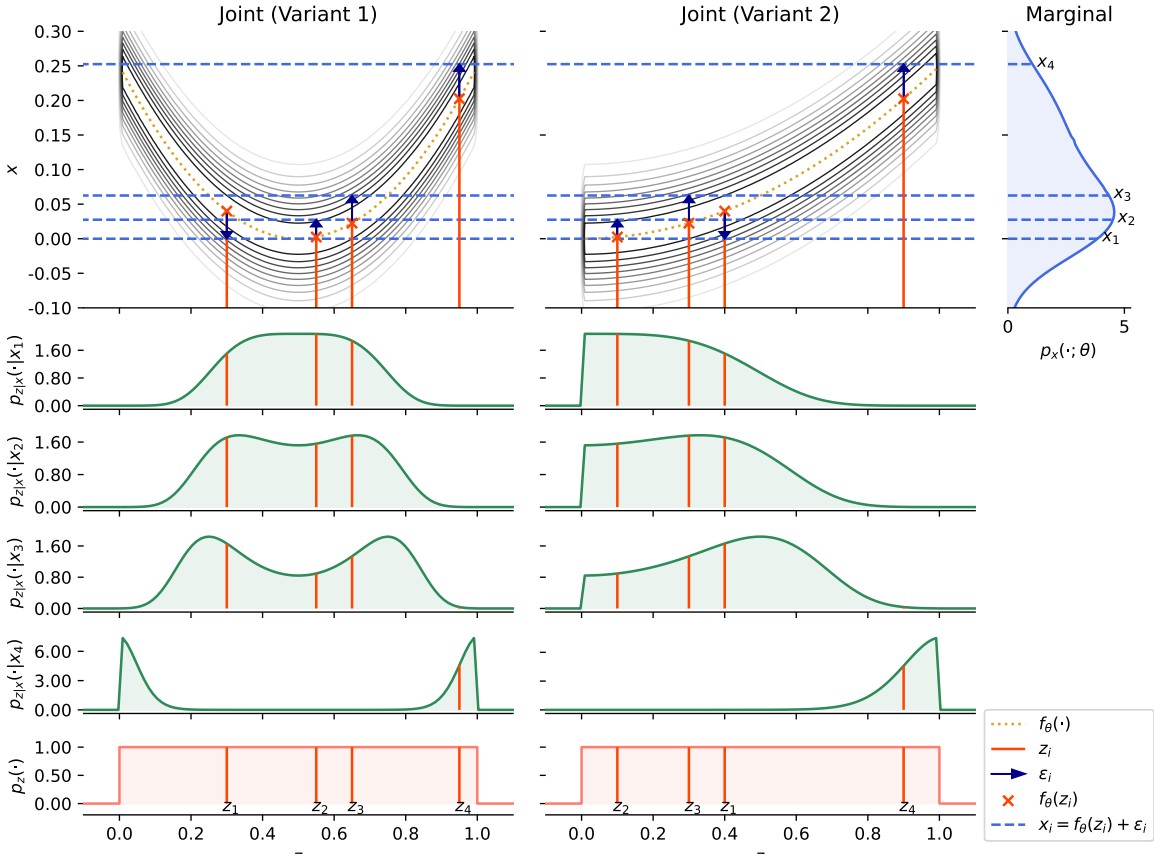

Figure 1: **Intuition behind MAPA: nearby points score highly under each other's posteriors—distant points do not.** Bottom row: four samples drawn from $p_z(\cdot) = \mathcal{U}[0,1]$. Top row: the generative process (i.e. $z_i$, $f_\theta(z_i)$, $\epsilon_i$, and the resultant $x_i$), visualized on-top of the joint distribution of the observed and latent variables (gray contours) for two different functions that yield the same marginal distribution (in blue). In "Variant 1," $f_\theta(z) = (0.5 - z)^2$, and in "Variant 2," $f_\theta(z) = 0.25 \cdot z^2$ (Yacoby et al., 2020b). The posteriors of each of the four observations $x_i$ are visualized in the green density plots.

We then derive the following stochastic lower bound (derivation in Appendix B.3):

$$\mathcal{L}_{\mathrm{MAPA}}^S(x_n; \theta, \psi) = \log\left( \frac{1}{N} \sum_{i \in \mathcal{B}_n(k)} p_{x|i,Z}(x_n | f_\theta \circ g_\psi(x_i)) + \frac{1}{NS} \sum_{s=1}^S \frac{p_{x|i,Z}(x_n | f_\theta \circ g_\psi(x_{i^{(s)}}))}{\tilde{q}_{i|n}^k(i^{(s)}|n)} \right),$$

(10)

where $i^{(s)} \sim \tilde{q}_{i|n}^k(i|n)$. As a reminder, we use the prior amortization scheme from Section 3 to represent $Z$ via $g_\psi(\cdot)$. When $k = 1$ and $S = 0$, we recover the AE loss, and increasing $k, S$

tightens the bound. After maximizing $\mathcal{L}_{\text{MAPA}}^{S}$ with respect to $\theta, \psi$, we learn a parametric prior distribution of $z_n = g_\psi(x_n)$ via method of our choice (e.g. KDE, Normalizing Flows).

**Generating samples.** After learning $\theta, \psi$ as described in this section, we can generate new samples using the *original* generative process from Eq. (1).

**Computational efficiency.** Computing $q_{i|n}(\cdot|n)$ requires pairwise comparisons $O(N^2 \cdot D)$, but can be parallelized. Since it only depends on the data-set, it can be computed once and shared online. Across random restarts and hyper-parameter selection, the cost of computing MAPA becomes advantageous. During training, we see a second boost in efficiency: $f_\theta \circ g_\psi$ is evaluated on at most $N$ points. We can therefore *reuse* forward passes when evaluating $\mathcal{L}_{\text{MAPA}}^{S}(\cdot; \theta, \psi)$ on a batch of points. As we show later, this will substantially reduce the number of forward passes needed per gradient computation, which will speed up training for models for which evaluations of $f_\theta \circ g_\psi$ dominate the computation.

## 6. Experiments and Results

**Setup.** We compare our approach (MAPA) with a VAE and IWAE with $p_z(\cdot; \psi) = \mathcal{N}(0, I)$ (fixed) on 5 synthetic data-sets (for which we know the ground-truth)—the "Figure-8," "Clusters," and "Spiral-Dots" examples for which mean-field Gaussian VAEs struggle, and the "Absolute-Value" and "Circle" examples for which they do not (Yacoby et al., 2020b) (details in Appendix C.1). Both the VAE and IWAE use a mean-field Gaussian $q_{z|x}(\cdot|x; \phi)$. We used 10 random restarts for each method (selecting the best random restart via validation log-likelihood (LL)), averaging results on 10 draws of each data-set. Each method was given the hyper-parameters of the ground-truth model (details in Appendix C.2). For all methods, LL was evaluated by (a) fixing the learned generative model parameters $\theta^*, \psi^*$ and fitting IWAE with a 50-component mixture of Gaussians $q_{z|x}(\cdot|x; \phi)$ with $S = 500$, and (b) approximating the LL with this bound with $S = 20000$. As such, our evaluation favors IWAE. For MAPA, we ensured that $p_z(\cdot; \psi)$ is Gaussian via the procedure in Appendix C.3.

**MAPA outperforms baselines on density estimation across different $S$.** MAPA outperforms the VAE and IWAE on density estimation across all but one of the data-sets. Further, MAPA performs as well with $q_{i|n}(\cdot|n)$ as it does with the true posterior of the approximate model ("MAPA-GT"). Lastly, when $q_{i|n}(\cdot|n)$ is artificially set to a uniform ("MAPA-naive"), it performs poorly, indicating that our proposed $q_{i|n}(\cdot|n)$ is necessary for good performance. See Fig. 2(a) (full results in Appendix D.1).

**MAPA inference requires fewer forward-passes.** We compare how many NN-passes MAPA requires vs. IWAE per gradient computation. Specifically, we plot the average number of NN-passes required when evaluating each method on a batch of size 100, varying the number of importance samples $S$. We find that, across all data-sets, when the cost of the decoder dominates the gradient computation, the cost of MAPA with $S = 200$ is that of IWAE's with $S = 50$. When the decoder *and* encoder dominate, the cost of MAPA with $S = 200$ is that of IWAE's with $S = 100$. See Fig. 2(b) (full results in Appendix D.2).

**MAPA captures trend of ground-truth posterior.** Across all data-sets, $q_{i|n}(\cdot|n)$ captures the trend of the ground-truth posterior of the empiricalized model (Eq. (7)), as well as of the original ground-truth model (full results in Appendix D.3).

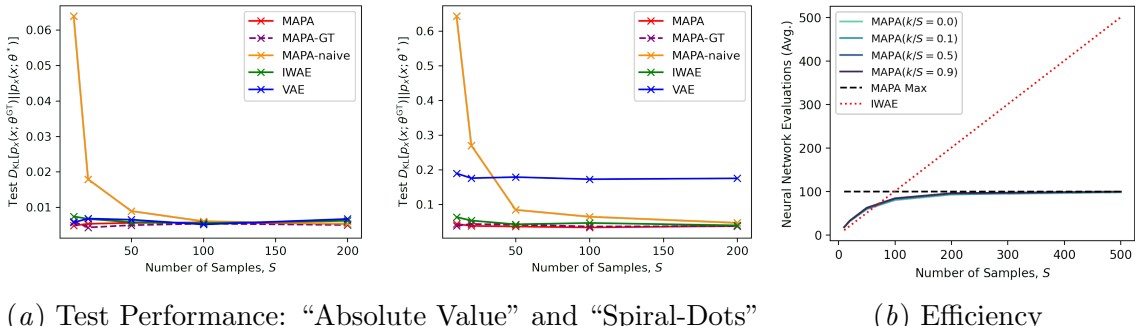

($a$) Test Performance: "Absolute Value" and "Spiral-Dots"   ($b$) Efficiency

Figure 2: (a) MAPA is more accurate than baselines; it better matches the true data distribution (lower test KL) than VAE and IWAE. (b) MAPA is more efficient; it requires fewer NN-passes (plots are similar for both data-sets). Full results and details in Appendices D.1 and D.2.

**MAPA is robust to model non-identifiability.** Given two different decoders $f_{\theta^{\text{GT}}}(\cdot) \neq f_{\hat\theta}(\cdot)$ that induce the same $p_x(\cdot; \theta^{\text{GT}}) = p_x(\cdot; \hat\theta)$, MAPA captures the trend in both equally well; it is therefore robust to model non-identifiability (full results in Appendix D.4).

## 7. Discussion and Future Work

We propose a novel, deterministic, model-agnostic posterior approximation and use it to develop a preliminary inference method for VAEs that is both accurate and faster than baselines: on low-dimensional synthetic data, our method requires fewer forward-passes and better captures the data distribution given a fixed computational budget.

**Roadmap to scaling MAPA-based inference to high-dimensions.** The preliminary inference method from Section 5 can be adapted to scale to higher dimensions in two ways. First, we can reduce the quadratic cost of estimating MAPA, e.g. via sketching algorithms, approximate nearest neighbor searches, or sparse KDE (thanks to its relationship with KDE). We can also reduce GPU-memory usage (from keeping MAPA in memory) by via batching schemes that reduce the memory footprint per batch. Second, Eq. (10) ignores the entropy due to the *location* of the latent codes—it should be adapted to account for this entropy for good performance in higher-dimensions (Welling et al., 2008).

**Theory.** In future work, we plan to theoretically analyze the tightness of our bound, as well as the distance of $q_{i|n}(\cdot|n)$ from the ground-truth posterior. We also plan to use sampling-without-replacement schemes in sampling from the proposal in Eq. (10) (e.g. Kool et al. (2019); Shi et al. (2020)), which will further tighten the bound to the LML.

**Extensions.** In this work, we only derive MAPA for a limited set of observation noise distributions. In future work, we plan to develop a more general method for specifying $\kappa(\cdot, \cdot)$ to allow additional observation noise models and data modalities (e.g. time-series), and to incorporate MAPA into other types of latent variable models.

## Acknowledgments

We thank Siddharth Swaroop for feedback on this manuscript. This material is based upon work supported by the National Science Foundation under Grant No. IIS-1750358. Any opinions, findings, and conclusions or recommendations expressed in this material are those of the author(s) and do not necessarily reflect the views of the National Science Foundation.

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

## Appendix Table of Contents

## Appendix A. Related Work

**Improving VAE inference.** Since VAE inference requires maximizing an intractable LML, recent work focuses on developing efficient and accurate approximate inference methods. The majority of this work introduces an inference model that is jointly optimized with the generative model to maximize a variational lower bound to the LML. As the inference model gets closer to the true posterior of the latent codes given the observations, the gap between the bound and the LML decreases. These bounds can therefore be tightened at an additional computational cost by increasing the flexibility of the inference model.

We loosely group these bounds into three categories of algorithms used to train latent variable models. First are algorithms that extend the classical Expectation-Maximization algorithm (EM) (Dempster et al., 1977) to Variational Inference; these algorithms optimize a single variational lower bound that can be tightened with more compute. For example, in the original formulation of the VAE (Kingma and Welling, 2014) and the $\alpha$-divergence formulation (Li and Turner, 2016), the bounds can be tightened by increasing the flexibility of the inference model (e.g. Rezende and Mohamed (2015); Yin and Zhou (2018)). In the importance weighted (or "divide-and-couple") formulation (e.g. Burda et al. (2016); Domke and Sheldon (2019)), the bound can be additionally tightened by increasing the number of samples in the inner MC-estimate. Similarly, in the thermodynamic formulation (Masrani et al., 2019), the bound can be tightened by increasing the number of "partitions" used. These bounds can be further combined in a variety of ways (e.g. Sobolev and Vetrov (2019); Daudel et al. (2023)). The second category of algorithms are those that extend the classical Wake-Sleep algorithm (Hinton et al., 1995); these algorithms use separate objectives to train the inference and generative models (Bornschein and Bengio, 2015; Várady et al., 2020; Le et al., 2020). The last category of algorithms are those that de-bias existing bounds via

classical techniques such as Jack-Knife (e.g. Quenouille (1949, 1956)) and Russian Roulette (Kahn, 1955) schemes (e.g. Nowozin (2018); Luo et al. (2020); Ishikawa and Goda (2021))).

In contrast to these works, our method does not optimize an inference model; instead, we propose a *deterministic* approximation to the *ground-truth* model's posterior, computed *once* per data-set, without needing to make assumptions about the ground-truth model. The works most similar to our approach are (a) Generative Topographic Mappings (Bishop et al., 1998), which also discretize the prior distribution (though they do this on a fixed grid over the latent space), and (b) Approximate Bayesian Computation (e.g. Sisson et al. (2018)), which use distances between observations generated from the prior (or original generative process) to estimate a posterior over unobserved variables.

**Mitigating non-identifiability in VAEs.** VAEs are known to be non-identifiable, in that their latent space can be transformed while still explaining the observed data equally well (e.g. Locatello et al. (2019); Yacoby et al. (2020a,b)). In such scenarios, it has been shown that the undesirable effects of non-identifiability can be mitigated by modifying the model itself to become identifiable (e.g. Khemakhem et al. (2020); Wang et al. (2021)), or by specifying additional model selection criteria (Zhao et al., 2018). In contrast to these works, we do not modify the original model to make it identifiable; instead, we propose an inference method that is agnostic to model non-identifiability, meaning that the inductive bias of the variational family does not affect the choice of learned model, allowing us to freely apply additional selection criteria to identify the model.

## Appendix B. Derivations

### B.1. Connections with (non-variational) autoencoders (AEs)

At a high-level, Eq. (6) can be seen as a generalization of an AE, in which we translate learning one empirical distribution (over the observation space) into learning an empirical distribution over another (the latent space). We make this connection concrete by showing that the AE loss is a lower bound to the LML of the empiricalized model in Eq. (6):

$$\log p_x(x_n; \theta, \psi) = \log \frac{1}{N} \sum_{i=1}^{N} p_{x|i,Z}(x_n | f_\theta \circ g_\psi(x_i)) \tag{11}$$

$$= \log \sum_{i=1}^{N} p_{x|i,Z}(x_n | f_\theta \circ g_\psi(x_i)) - \log N \tag{12}$$

$$= \log \left( p_{x|i,Z}(x_n | f_\theta \circ g_\psi(x_n)) + \sum_{i \neq n} p_{x|i,Z}(x_n | f_\theta \circ g_\psi(x_i)) \right) - \log N \tag{13}$$

$$= \underbrace{\log p_{x|i,Z}(x_n | f_\theta \circ g_\psi(x_n)) - \log N}_{\text{autoencoder loss + const.}} + \underbrace{\log \left( 1 + \frac{\sum_{i \neq n} p_{x|i,Z}(x_n | f_\theta \circ g_\psi(x_i))}{p_{x|i,Z}(x_n | f_\theta \circ g_\psi(x_n))} \right)}_{\text{"gap" or "regularizer"} \geq 0} \tag{14}$$

In the above, the first term is the AE reconstruction objective, and the second term is non-zero (it's the log of a value $\geq 1$), representing a "gap" / "regularizer." For large $N$, the AE loss is also a lower bound to the original LML. This decomposition is different than the popular ELBO decomposition into a reconstruction term (often regarded as analogous to the AE's objective) and an information-theoretic regularizer (KL-divergence between the variational posterior and prior).

## B.2. Derivation of MAPA

To see why $q_{i|n}(\cdot|n)$, defined in Eq. (8), is a sensible approximation, we relate it to the true posterior of the empiricalized model. For the following derivation, we assume a Gaussian observation noise; that is, we assume that $x_n = f(z_n; \theta) + \epsilon_n$, where $\epsilon_n \sim \mathcal{N}(0, \sigma_\epsilon^2 \cdot I)$. We further assume that we observed $\theta^{\text{GT}}$ and $Z^{\text{GT}}$, as well as the ground-truth values of the noise, $\epsilon_n^{\text{GT}}$. We begin our derivation by relying on all of these components of the ground-truth data-generating process, $\theta^{\text{GT}}, Z^{\text{GT}}, \epsilon_n^{\text{GT}}$, and we'll end up with an analytic form that depends on none of them:

$$p_{i|x,Z}(i|x_n; \theta^{\text{GT}}, Z^{\text{GT}}) = \frac{p_i(i) \cdot p_{x|i,Z}(x_n|f_{\theta^{\text{GT}}}(z_i^{\text{GT}}))}{\sum\limits_{j=1}^{N} p_i(j) \cdot p_{x|i,Z}(x_n|f_{\theta^{\text{GT}}}(z_j^{\text{GT}}))} \tag{15}$$

$$= \frac{p_{x|i,Z}(x_n|f_{\theta^{\text{GT}}}(z_i^{\text{GT}}))}{\sum\limits_{j=1}^{N} p_{x|i,Z}(x_n|f_{\theta^{\text{GT}}}(z_j^{\text{GT}}))} \tag{16}$$

$$\approx \frac{p_{x|i,Z}(x_n|f_{\theta^{\text{GT}}}(z_i^{\text{GT}}) + \epsilon_i^{\text{GT}})}{\sum\limits_{j=1}^{N} p_{x|i,Z}(x_n|f_{\theta^{\text{GT}}}(z_j^{\text{GT}}) + \epsilon_j^{\text{GT}})} \quad \text{since} \quad x_j = f_{\theta^{\text{GT}}}(z_j^{\text{GT}}) + \epsilon_j^{\text{GT}}$$

$$\tag{17}$$

$$= \frac{\kappa(x_n|x_i)}{\sum\limits_{j=1}^{N} \kappa(x_n|x_j)} \tag{18}$$

$$= q_{i|n}(i|n) \tag{19}$$

where $\kappa(x_n|x_i)$ is a Gaussian (RBF) kernel with bandwidth $\sigma_\epsilon^2$.

Similar derivations hold for likelihood distributions whose support is the same as the support of their parameters (e.g. the Continuous Bernoulli (Loaiza-Ganem and Cunningham, 2019)). For distributions for which this property does not hold, we have to tweak the derivation. For example, for a Bernoulli likelihood, a naive application of MAPA will yield,

$$\kappa(x_n|x_i) = \prod_{d=1}^{D} x_i^{(d)} \cdot x_n^{(d)} + (1 - x_i^{(d)}) \cdot (1 - x_n^{(d)}), \tag{20}$$

which will be 0 if $x_n$ and $x_i$ do not match in even one location $d$. As such, we tweak the above by softening the probabilities with a hyper-parameter $\rho \in (0, 1)$:

$$\kappa(x_n|x_i; \rho) = \prod_{d=1}^{D} (\rho \cdot x_i^{(d)}) \cdot x_n^{(d)} + (1 - \rho \cdot x_i^{(d)}) \cdot (1 - x_n^{(d)}), \tag{21}$$

selected to be close to 1 (e.g. $\rho = 0.9$). Here, $\rho$ controls the "peakiness" of the posterior approximation.

In both the Gaussian and Bernoulli cases, notice that there's a hyper-parameter that needs to be selected ($\sigma_\epsilon^2$ and $\rho$, respectively). In both cases, the bulk of the computation is in computing pairwise differences between all points (using different notions of distance, depending on the case). Given a matrix of pairwise differences, one can apply these hyper-parameters post-hoc, adding little overhead.

## B.3. Derivation of the MAPA-based stochastic lower bound

We leverage $q_{i|n}(\cdot|n)$ to derive a lower bound to the LML of the empiricalized model from Eq. (6):

$$\log p_x(x_n; \theta, \psi) = \log \frac{1}{N} \sum_{i=1}^{N} p_{x|i,Z}(x_n | f_\theta \circ g_\psi(x_i)) \tag{22}$$

$$= \log \sum_{i=1}^{N} p_{x|i,Z}(x_n | f_\theta \circ g_\psi(x_i)) - \log N. \tag{23}$$

We define $\mathcal{B}_n(k)$ to be the set of $k$ indices for which $q_{i|n}(\cdot|n)$ is largest, where $k \in \{1, \ldots, N\}$. We use membership in this set to split the above sum into two sums:

$$\log p_x(x_n; \theta, \psi) = \log \left( \sum_{i \in \mathcal{B}_n(k)} p_{x|i,Z}(x_n | f_\theta \circ g_\psi(x_i)) + \sum_{i \notin \mathcal{B}_n(k)} p_{x|i,Z}(x_n | f_\theta \circ g_\psi(x_i)) \right) - \log N, \tag{24}$$

where the second term will be approximated with a $S$-sample importance weighted lower bound. We split the objective into two sums for two reasons. First, this maintains a clear connection with (non-variational) AEs; when $k = 1, S = 0$, this objective reduces to the AE loss. Second, when $q_{i|n}(\cdot|n)$ has a long tail, we expect increasing $k$ would reduce variance. In essence, this objective uses a nearest-neighbor approximation of the expectations for the empiricalized model's LML.

Next, we approximate the gap via a stochastic lower bound. To do this, we define, $\tilde{q}_{i|n}^k(i|n)$ to be $q_{i|n}(\cdot|n)$ renormalized after setting the probability of its $k$ largest elements to 0:

$$\tilde{q}_{i|n}^k(i|n) = \frac{q_{i|n}(i|n) \cdot \mathbb{I}[i \notin \mathcal{B}_n(k)]}{\sum_{j=1}^{N} q_{i|n}(j|n) \cdot \mathbb{I}[j \notin \mathcal{B}_n(k)]}, \tag{25}$$

We now approximate the second term inside the log as follows:

$$\sum_{i \notin \mathcal{B}_n(k)} p_{x|i,Z}(x_n | f_\theta \circ g_\psi(x_i)) = \sum_{i \notin \mathcal{B}_n(k)} \tilde{q}_{i|n}^k(i|n) \cdot \frac{p_{x|i,Z}(x_n | f_\theta \circ g_\psi(x_i))}{\tilde{q}_{i|n}^k(i|n)} \tag{26}$$

$$= \mathbb{E}_{i \sim \tilde{q}_{i|n}^k(i|n)} \left[ \frac{p_{x|i,Z}(x_n | f_\theta \circ g_\psi(x_i))}{\tilde{q}_{i|n}^k(i|n)} \right] \tag{27}$$

$$\approx \frac{1}{S} \sum_{s=1}^{S} \frac{p_{x|i,Z}(x_n | f_\theta \circ g_\psi(x_{i^{(s)}}))}{\tilde{q}_{i|n}^k(i^{(s)}|n)}, \quad i^{(s)} \sim \tilde{q}_{i|n}^k(i|n) \tag{28}$$

This gives us the following importance weighted stochastic lower bound:

$$\mathcal{L}_{\text{MAPA}}^S(x_n; \theta, \psi) = \log \left( \sum_{i \in \mathcal{B}_n(k)} p_{x|i,Z}(x_n | f_\theta \circ g_\psi(x_i)) + \frac{1}{S} \sum_{s=1}^{S} \frac{p_{x|i,Z}(x_n | f_\theta \circ g_\psi(x_{i^{(s)}}))}{\tilde{q}_{i|n}^k(i^{(s)}|n)} \right) - \log N, \tag{29}$$

where $i^{(s)} \sim \tilde{q}_{i|n}^k(i|n)$. That is,

$$\log p_x(x_n; \theta, \psi) \geq \mathbb{E}_{i^{(1)}, \dots, i^{(S)} \sim \tilde{q}_{i|n}^k(i|n)} \left[ \mathcal{L}_{\text{MAPA}}^S(x_n; \theta, \psi) \right]. \tag{30}$$

Like the IWAE-bound, this bound tightens as $S$ or $k$ increase. Unlike the IWAE-bound, however, this bound does not require specialized gradient estimators, since it does not differentiate with respect to $q_{i|n}(\cdot|n)$.

## Appendix C. Experimental Setup

### C.1. Data

In this section, we describe the synthetic examples used in this paper. We chose these data-sets because they have been previously used to demonstrate pathologies of VAE inference (Yacoby et al., 2020b). For each one of these example decoder functions, we fit a surrogate NN, $f_\theta$, with 3 layers of 50 hidden nodes using full supervision (ensuring that the MSE $< 1e-4$ and use that $f_\theta$ to generate the actual data used in the experiments. For each data-set, we generated $N = 5000$ points.

**Figure-8 Example.** Let $\Phi(z)$ is the Gaussian CDF and $\sigma_\epsilon^2 = 0.02$.

$$\begin{aligned}
z &\sim \mathcal{N}(0, 1) \\
\epsilon &\sim \mathcal{N}(0, \sigma_\epsilon^2 \cdot I) \\
u(z) &= (0.6 + 1.8 \cdot \Phi(z)) \pi \\
x|z &= \underbrace{\begin{bmatrix} \frac{\sqrt{2}}{2} \cdot \frac{\cos(u(z))}{\sin(u(z))^2 + 1} \\ \sqrt{2} \cdot \frac{\cos(u(z)) \sin(u(z))}{\sin(u(z))^2 + 1} \end{bmatrix}}_{f_{\theta \text{GT}}(z)} + \epsilon
\end{aligned} \tag{31}$$

**Circle Example.** Let $\Phi(z)$ is the Gaussian CDF and $\sigma_\epsilon^2 = 0.01$.

$$
\begin{aligned}
z &\sim \mathcal{N}(0, 1) \\
\epsilon &\sim \mathcal{N}(0, \sigma_\epsilon^2 \cdot I) \\
x|z &= \underbrace{\begin{bmatrix} \cos(2\pi \cdot \Phi(z)) \\ \sin(2\pi \cdot \Phi(z)) \end{bmatrix}}_{f_{\theta \mathrm{GT}}(z)} + \epsilon
\end{aligned}
\tag{32}
$$

**Absolute-Value Example.** Let $\Phi(z)$ is the Gaussian CDF and $\sigma_\epsilon^2 = 0.01$.

$$
\begin{aligned}
z &\sim \mathcal{N}(0, 1) \\
\epsilon &\sim \mathcal{N}(0, \sigma_\epsilon^2 \cdot I) \\
x|z &= \underbrace{\begin{bmatrix} |\Phi(z)| \\ |\Phi(z)| \end{bmatrix}}_{f_{\theta \mathrm{GT}}(z)} + \epsilon
\end{aligned}
\tag{33}
$$

**Clusters Example.** Let $\sigma_\epsilon^2 = 0.2$.

$$
\begin{aligned}
z &\sim \mathcal{N}(0, 1) \\
\epsilon &\sim \mathcal{N}(0, \sigma_\epsilon^2 \cdot I) \\
u(z) &= \frac{2\pi}{1 + e^{-\frac{1}{2}\pi z}} \\
t(u) &= 2 \cdot \tanh\left(10 \cdot u - 20 \cdot \lfloor u/2 \rfloor - 10\right) + 4 \cdot \lfloor u/2 \rfloor + 2 \\
x|z &= \underbrace{\begin{bmatrix} \cos(t(u(z))) \\ \sin(t(u(z))) \end{bmatrix}}_{f_{\theta \mathrm{GT}}(z)} + \epsilon
\end{aligned}
\tag{34}
$$

**Spiral-Dots Example.** Let $\sigma_\epsilon^2 = 0.01$.

$$
\begin{aligned}
z &\sim \mathcal{N}(0, 1) \\
\epsilon &\sim \mathcal{N}(0, \sigma_\epsilon^2 \cdot I) \\
u(z) &= \frac{4\pi}{1 + e^{-\frac{1}{2}\pi z}} \\
t(u) &= \tanh\left(10 \cdot u - 20 \cdot \lfloor u/2 \rfloor - 10\right) + 2 \cdot \lfloor u/2 \rfloor + 1 \\
x|z &= \underbrace{\begin{bmatrix} t(u(z)) \cdot \cos(t(u(z))) \\ t(u(z)) \cdot \sin(t(u(z))) \end{bmatrix}}_{f_{\theta \mathrm{GT}}(z)} + \epsilon
\end{aligned}
\tag{35}
$$

### C.2. Hyper-parameters

Across all synthetic data, we fix the hyper-parameters that match those of the ground-truth data-generating process. Specifically, we fix the latent dimensions $L$, observation noise variance $\sigma_\epsilon^2$, and architecture of the NNs to those of the ground-truth (see Appendix C.1) for details. We selected the remaining hyper-parameters using validation log-likelihood.

**Optimization.** To train each model, we used the Adam optimizer (Kingma and Ba, 2015) with a learning rate of 0.001, and batch size of 100 for 500 epochs. To train the IWAE-bound used for evaluation (with the mixture of Gaussians $q_{z|x}(\cdot|x; \phi)$, described in the main text), we use a learning rate of 0.001 and a batch size of 1000 for 100 epochs. Lastly, to learn the MAPA prior (Appendix C.3), we used a learning rate of 0.0005, a batch size of 100 or 5000, for 500 epochs.

**MAPA.** We selected $k$ (in Eq. (10)) to be either 10% or 90% of the number of importance samples in the bound $S$.

### C.3. Copula-based prior recovery for 1D latent spaces

Given $\theta^*, \phi^*$ learned by maximizing $\mathcal{L}^S_{\text{MAPA}}(x_n; \theta, \psi)$, we have to learn a parametric form for the distribution of all $z_n = g_{\psi^*}(x_n)$. Since our synthetic data-sets are in one dimension and since we found Normalizing Flow training to be finicky, we use the following procedure to ensure the resultant prior is a standard Gaussian. This helped us ensure our evaluation was of our proposed bound only, and is not hindered by Normalizing Flow optimization. We emphasize that this process only works when the latent space is 1D, and that we only chose this procedure for its reliability in evaluating our proposed method.

1. Compute $z_n = g_{\psi^*}(x_n)$ for all $n$.

2. Compute the empirical Gaussian copula of the data:

$$u_n = \Phi^{-1}\left( \frac{1}{N} \sum_{i=1}^{N} \mathbb{I}(z_i \geq z_n) \right), \tag{36}$$

   where $\Phi^{-1}$ is the inverse CDF of a standard Gaussian.

3. Whiten the resultant Gaussian:

$$z_n^* = \frac{u_n - \mu_u}{\sigma_u}, \tag{37}$$

   where $\mu_u, \sigma_u$ are the sample mean and standard deviation. At this point, $z_n^*$ should be distributed like a standard Normal, which is our desired prior.

4. Now that we have transformed our original latent space into a Gaussian, all that's left is learning a function to map to and from this Gaussian as follows:

$$f_{\theta^*} \circ g_{\psi^*}(\cdot) = \underbrace{f_{\theta^*} \circ h}_{f_{\theta\dagger}} \circ \underbrace{h^{-1} \circ g_{\psi^*}}_{g_{\psi\dagger}}(\cdot). \tag{38}$$

   We do this by solving the following optimization problem:

$$\phi^\dagger = \text{argmin}_\psi \frac{1}{N} \sum_{n=1}^{N} \|z_n^* - g_\phi(x_n)\|_2^2, \tag{39}$$

$$\theta^\dagger = \text{argmin}_\theta \frac{1}{N} \sum_{n=1}^{N} D_{\text{KL}} \left[ p_{x|z}(x_n | f_{\theta^*} \circ g_{\psi^*}(x_n)) || p_{x|z}(x_n | f_\theta \circ g_{\psi\dagger}(x_n)) \right]. \tag{40}$$

## Appendix D. Results

### D.1. MAPA better estimates density across different $S$

In Fig. 3, we compare MAPA's performance in estimating the observed data distribution relative to baselines. We do this by computing $D_{\mathrm{KL}}\left[p_x(\cdot; \theta^{\mathrm{GT}}, \psi^{\mathrm{GT}}) || p_x(\cdot; \theta^*, \psi^*)\right]$, where $p_x(\cdot; \theta^{\mathrm{GT}}, \psi^{\mathrm{GT}})$ refers to the ground-truth model (see Appendix C.1) and $p_x(\cdot; \theta^*, \psi^*)$ refers to the learned model (all with the prior fixed to a standard Gaussian). Fig. 3 shows that, except on the "Clusters" Example, for which the MAPA is less accurate, MAPA outperforms both the VAE and IWAE on density estimation; it achieves a lower test KL. Further, MAPA performs as well with $q_{i|n}(\cdot|n)$ as it does with the true posterior of the approximate model ("MAPA-GT"), defined in Eq. (7). Lastly, when $q_{i|n}(\cdot|n)$ is artificially set to a uniform ("MAPA-naive"), it performs poorly, indicating that our model-agnostic posterior approximation is indeed effective.

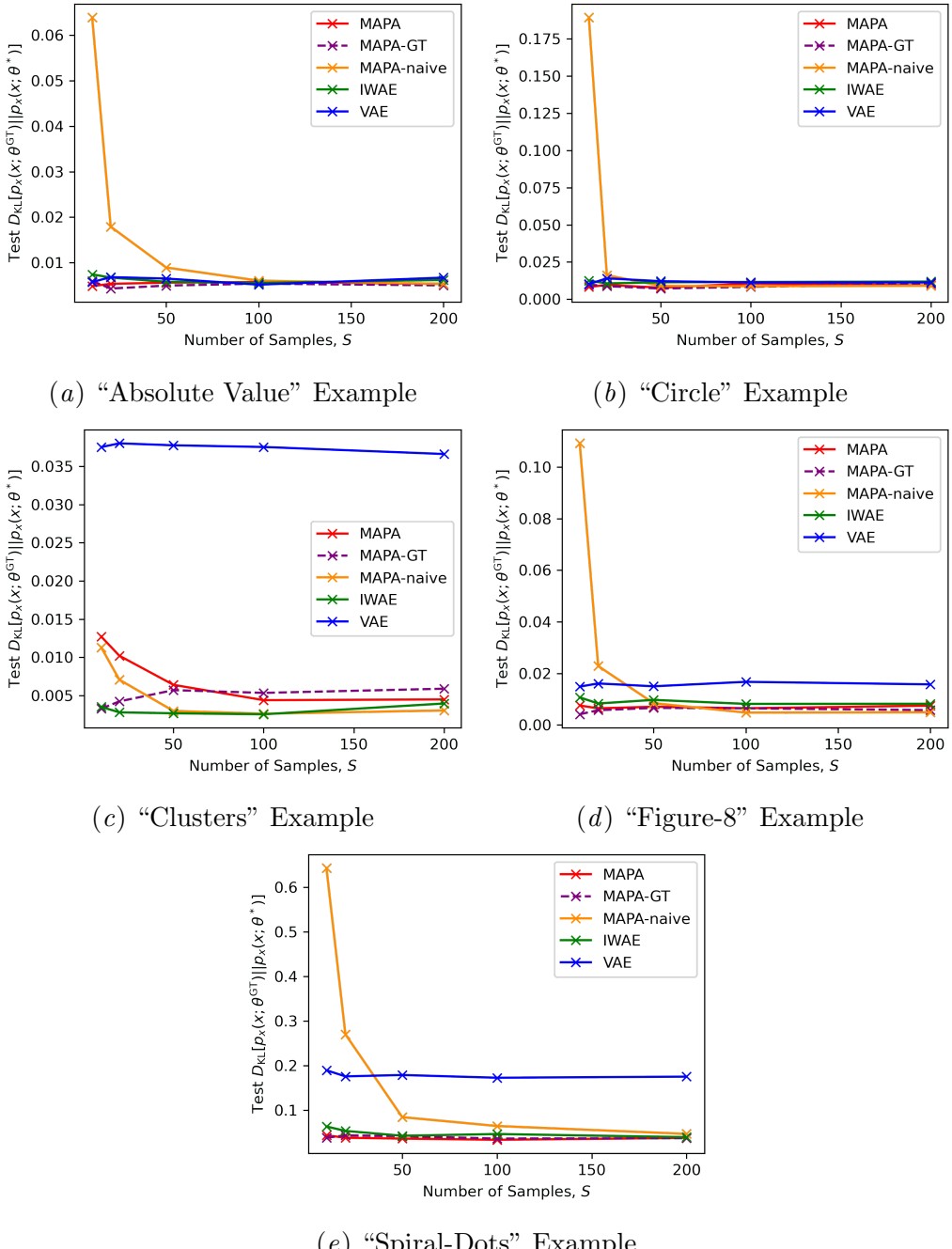

$(a)$ "Absolute Value" Example

$(b)$ "Circle" Example

$(c)$ "Clusters" Example

$(d)$ "Figure-8" Example

$(e)$ "Spiral-Dots" Example

Figure 3: **MAPA better estimates density across different $S$.** Except for on the "Clusters" Example, for which the MAPA is less accurate (see Fig. 8), MAPA outperforms baselines on density estimation (i.e. achieves a lower test $D_{\mathrm{KL}}\left[p_x(\cdot; \theta^{\mathrm{GT}}, \psi^{\mathrm{GT}})||p_x(\cdot; \theta^*, \psi^*)\right]$). Further, MAPA performs as well with $q_{i|n}(\cdot|n)$ as it does with the true posterior of the approximate model ("MAPA-GT"). Lastly, when $q_{i|n}(\cdot|n)$ is artificially set to a uniform ("MAPA-naive"), it performs poorly, indicating that the posterior approximation is what explains the good performance.

### D.2. MAPA inference requires fewer forward-passes

Figs. 4 and 5 compares how many NN-passes MAPA requires vs. IWAE per gradient computation. Specifically, we plot the average number of NN-passes required when evaluating each method on a batch of size 100, varying the number of importance samples $S$. In Fig. 4, we assume that the cost of the decoder NN dominates the computation of the objective, whereas in Fig. 5, we assume that the cost of the decoder and encoder NNs equally dominate the computation. "MAPA Max" is the maximum number of samples that MAPA can use (the number of data points $N$). For readability, we divide the number of forward passes by the batch size to get the number of forward passes needed per point.

Both figures show that MAPA requires significantly fewer forward passes than IWAE. We find that, across all data-sets, when the cost of the decoder dominates the gradient computation, the cost of MAPA with $S = 200$ is roughly that of IWAE's with $S = 50$ (Fig. 4). Similarly, when the decoder *and* encoder dominate, the cost of MAPA with $S = 200$ is roughly that of IWAE's with $S = 100$ (Fig. 5). This result potentially makes MAPA more memory efficient and thus better suited for GPUs (though we have not tested this).

Lastly, while the bounds tighten as $k$ increases, both figures show that the additional number of forward passes is negligible.

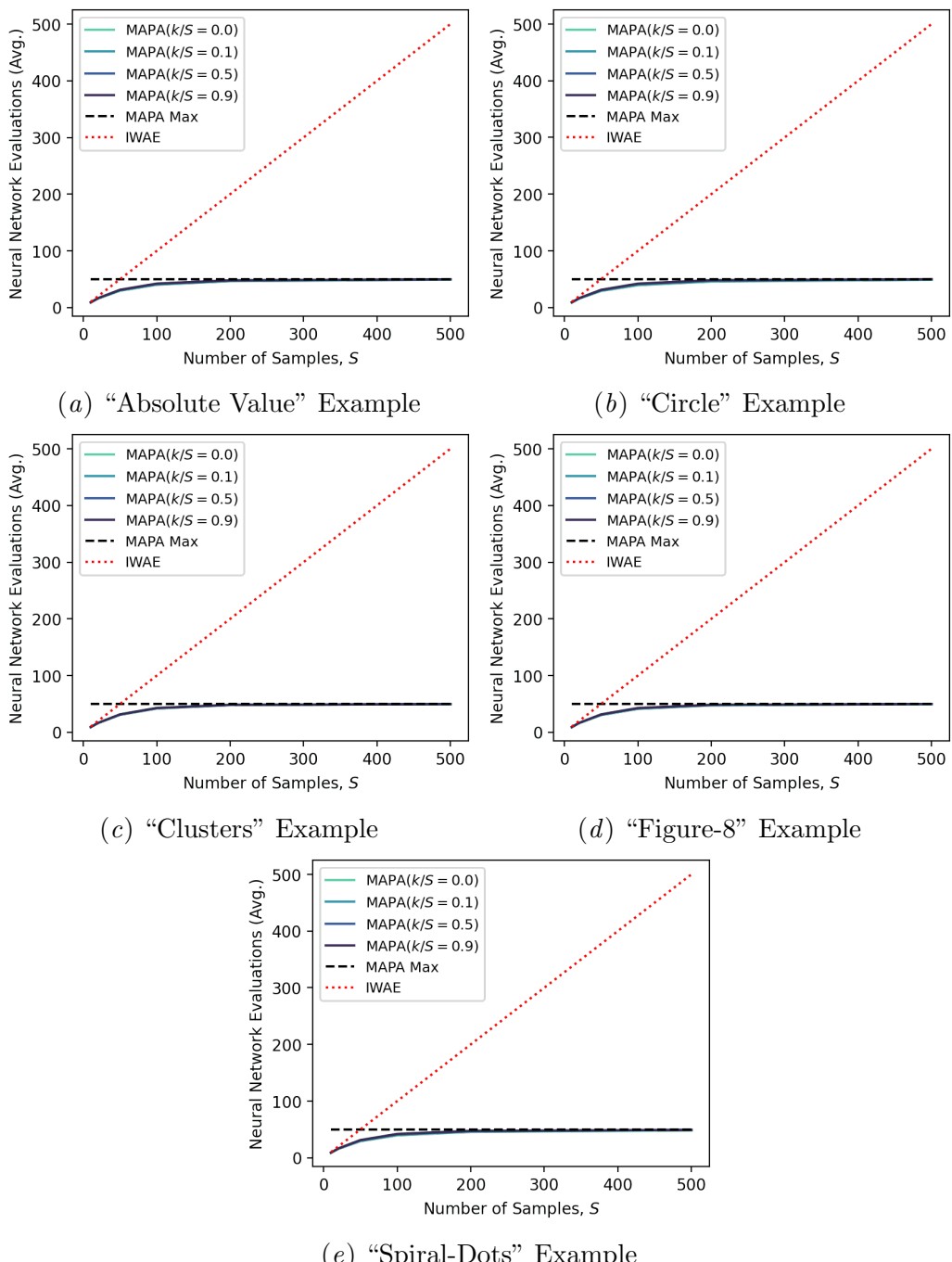

(a) "Absolute Value" Example

(b) "Circle" Example

(c) "Clusters" Example

(d) "Figure-8" Example

(e) "Spiral-Dots" Example

Figure 4: **MAPA requires fewer forward-passes than IWAE.** Here, we assume that the forward pass through the decoder is significantly more expensive than that of the encoder of both MAPA and IWAE, and that it dominates the rest of the computation of the objective. We plot the average number of NN evaluations required (per iteration of gradient descent, per point) given a batch size of 100.

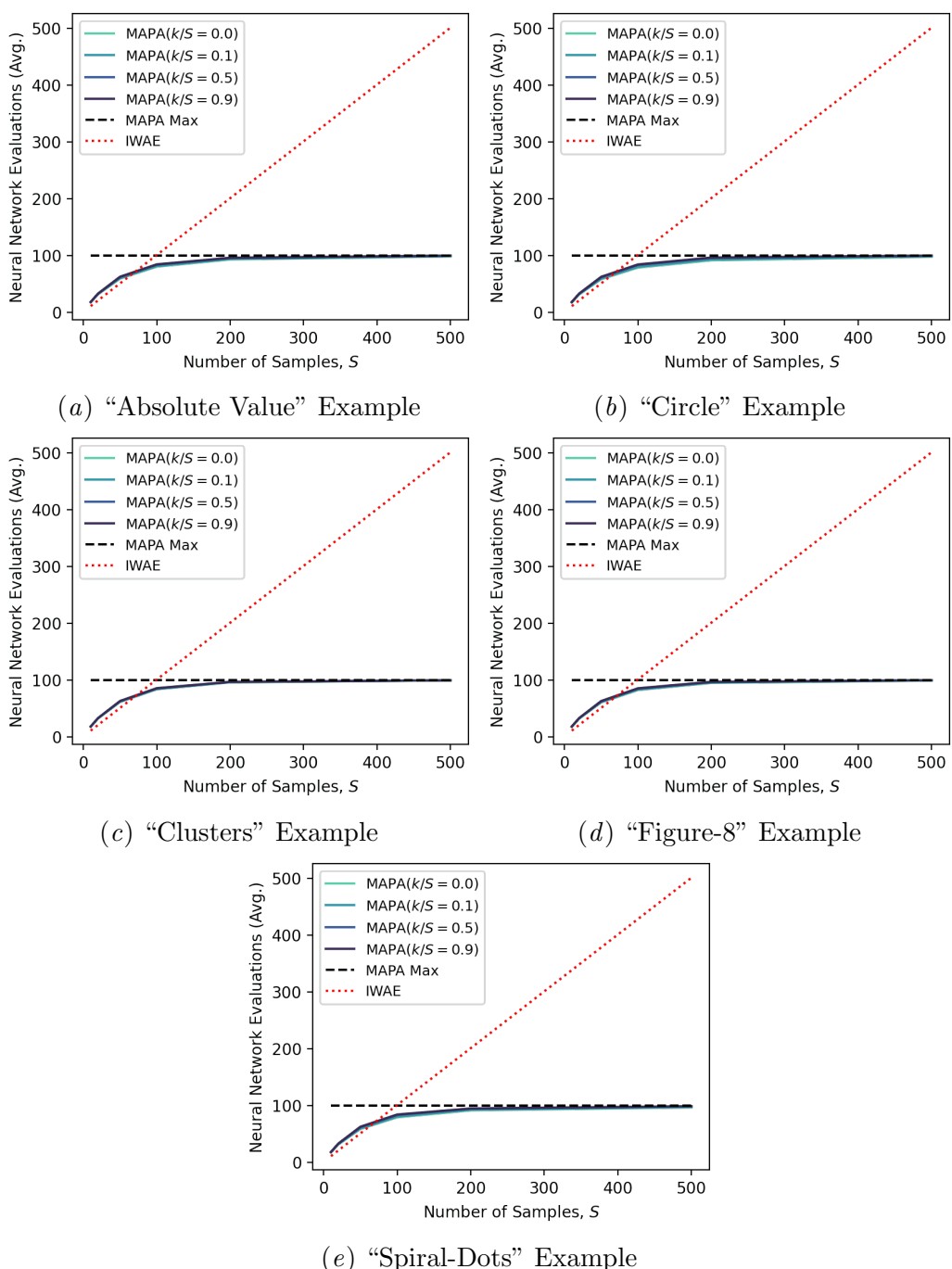

(a) "Absolute Value" Example

(b) "Circle" Example

(c) "Clusters" Example

(d) "Figure-8" Example

(e) "Spiral-Dots" Example

Figure 5: **MAPA requires fewer forward-passes than IWAE.** Here, we assume that the forward pass through the decoder and encoder networks of both MAPA and IWAE are equally expensive, and dominate the rest of the computation of the objective. We plot the average number of NN evaluations required (per iteration of gradient descent, per point) given a batch size of 100.

### D.3. MAPA captures trend of ground-truth posterior

Across different values of $x$, Figs. 6 to 10 compare:

1. The true posterior of the ground-truth model (black):

$$p_{z|x}(z|x; \theta^{\mathrm{GT}}, \psi^{\mathrm{GT}}) = \frac{p_{x|z}(x|f_{\theta^{\mathrm{GT}}}(z)) \cdot p_z(z; \psi^{\mathrm{GT}})}{p_x(x; \theta^{\mathrm{GT}}, \psi^{\mathrm{GT}})}. \tag{41}$$

2. The true posterior of the ground-truth empiricalized model (red), defined in Eq. (7): $p_{i|x,Z}(i|x; \theta^{\mathrm{GT}}, Z^{\mathrm{GT}})$.

3. MAPA (blue), defined in Eq. (8): $q_{i|n}(i|n)$.

The figures plot the above log posteriors relative to the ground-truth latent codes $z^{\mathrm{GT}}$. Note: since $p_{i|x,Z}(i|x; \theta^{\mathrm{GT}}, Z^{\mathrm{GT}})$ and $q_{i|n}(i|n)$ both have sums (as opposed integrals) in their denominator, they require scaling by $N$ to be plotted with the same units as $p_{z|x}(z|x; \theta^{\mathrm{GT}}, \psi^{\mathrm{GT}})$.

The figures all show that the $p_{z|x}(z|x; \theta^{\mathrm{GT}}, \psi^{\mathrm{GT}})$ matches $p_{z|x}(z|x; \theta^{\mathrm{GT}}, \psi^{\mathrm{GT}})$ (red matches black) nearly perfectly. The only places the two may diverge is towards the extreme values of $z^{\mathrm{GT}}$. This is because for these values of $z^{\mathrm{GT}}$, the prior is low, which is reflected in the density of the red points in the plot, not in the empiricalized prior. As such, they actually reflect the same trends.

The figures further show that $q_{i|n}(i|n)$ captures the trend of $p_{i|x,Z}(i|x; \theta^{\mathrm{GT}}, Z^{\mathrm{GT}})$; the blue points look like "noise" centered around the red curve. This noise comes from the addition of $\epsilon^{\mathrm{GT}}$ in the derivation of MAPA (Appendix B.2). Since our bound uses $q_{i|n}(i|n)$ as an importance sampling distribution, it suffices to capture the trend for a tight bound. The only data-set for which this "noise" presents an issue is the "Clusters" data-set (Fig. 8), suggesting MAPA may not work as well on this type of data-set.

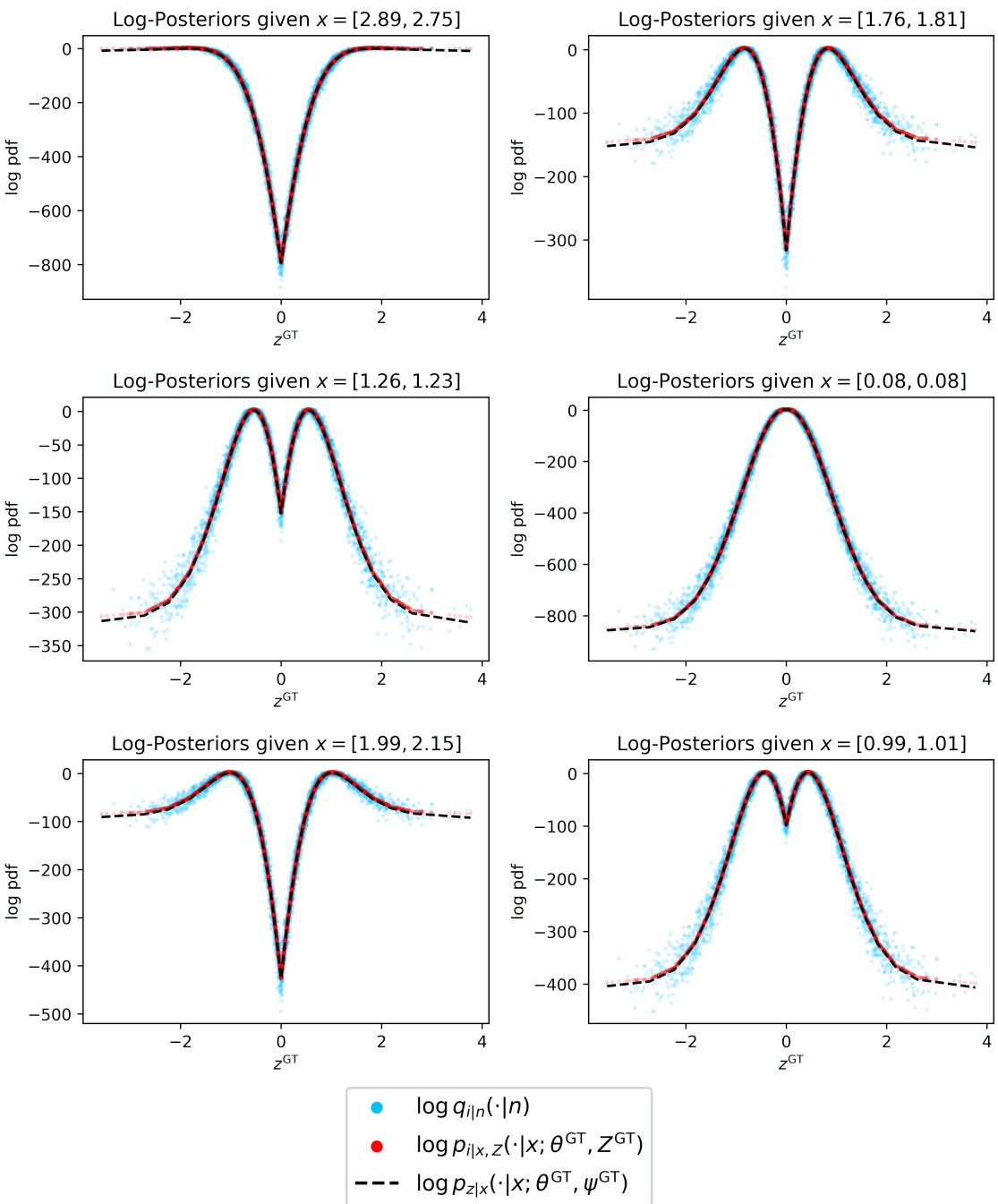

Figure 6: **MAPA captures trend of ground-truth posterior on "Absolute-Value" Example.** The panels compare the log-posteriors (of different $x$'s) vs. $z^{\text{GT}}$. The black, red and blue represent the true posterior of the original model, the true posterior of the empiricalized model, and the MAPA, respectively. Details in Appendix D.3.

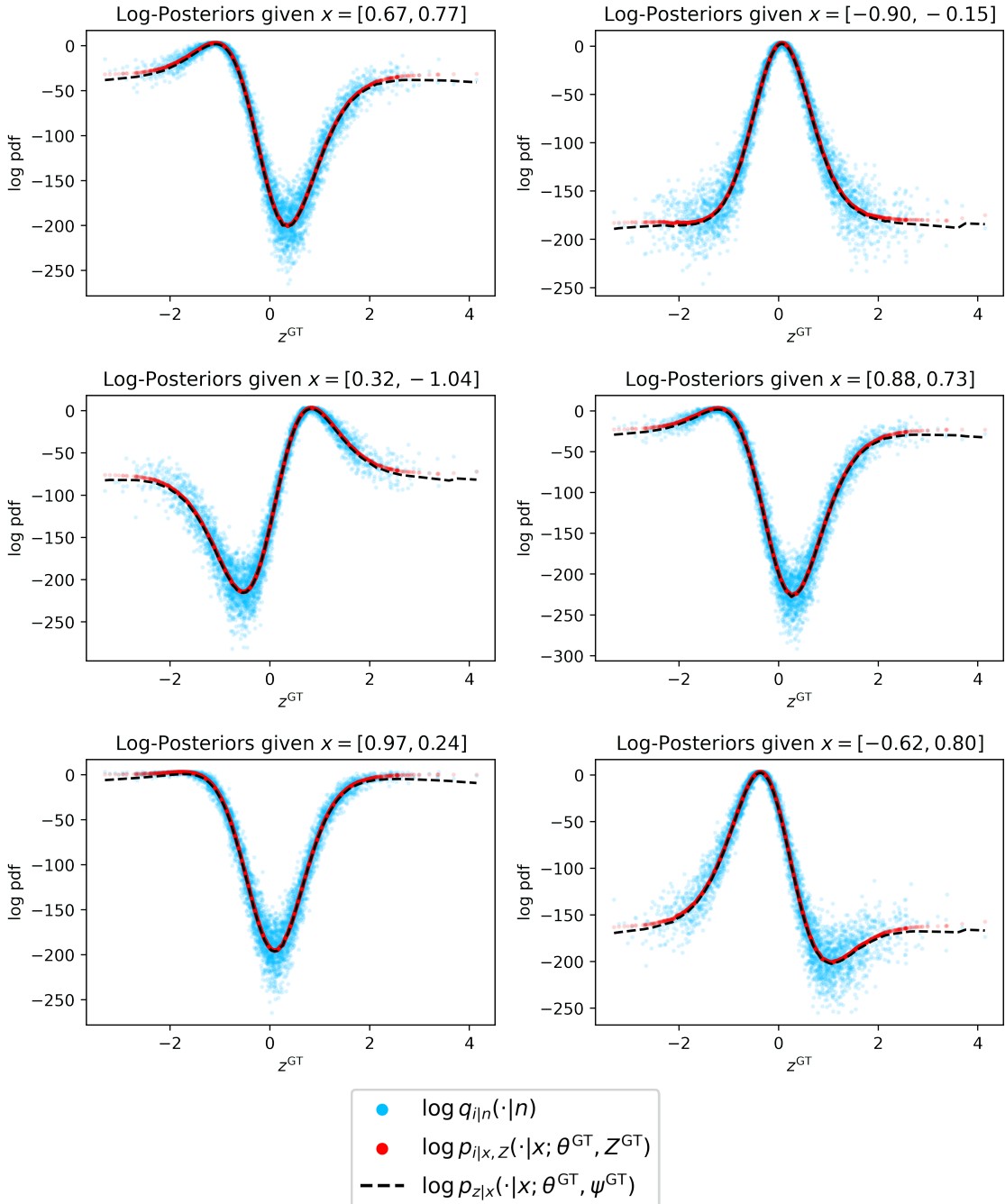

Figure 7: **MAPA captures trend of ground-truth posterior on "Circle" Example.** The panels compare the log-posteriors (of different $x$'s) vs. $z^{\mathrm{GT}}$. The black, red and blue represent the true posterior of the original model, the true posterior of the empiricalized model, and the MAPA, respectively. Details in Appendix D.3.

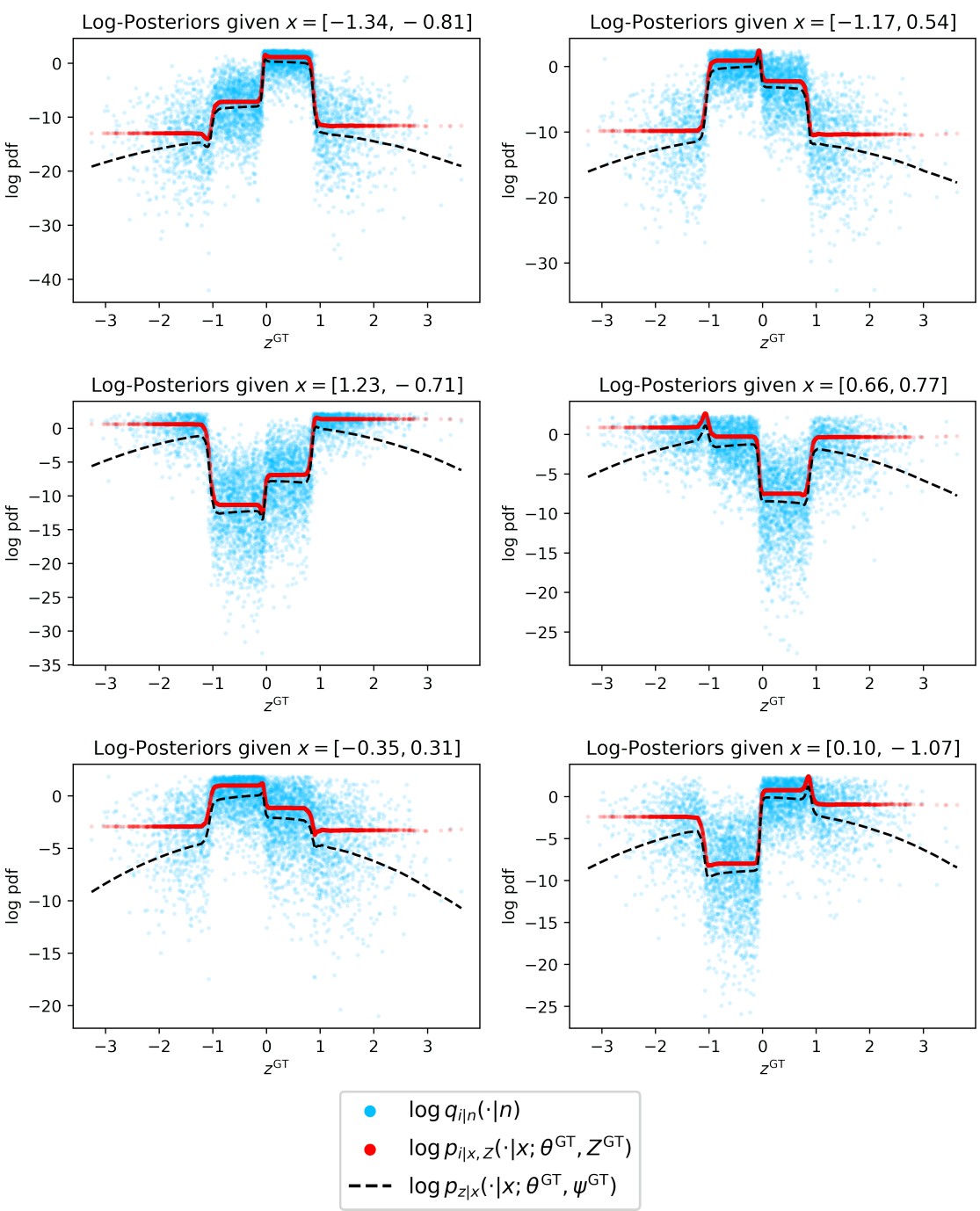

Figure 8: **MAPA captures trend of ground-truth posterior on "Clusters" Example.** The panels compare the log-posteriors (of different $x$'s) vs. $z^{\mathrm{GT}}$. The black, red and blue represent the true posterior of the original model, the true posterior of the empiricalized model, and the MAPA, respectively. Details in Appendix D.3.

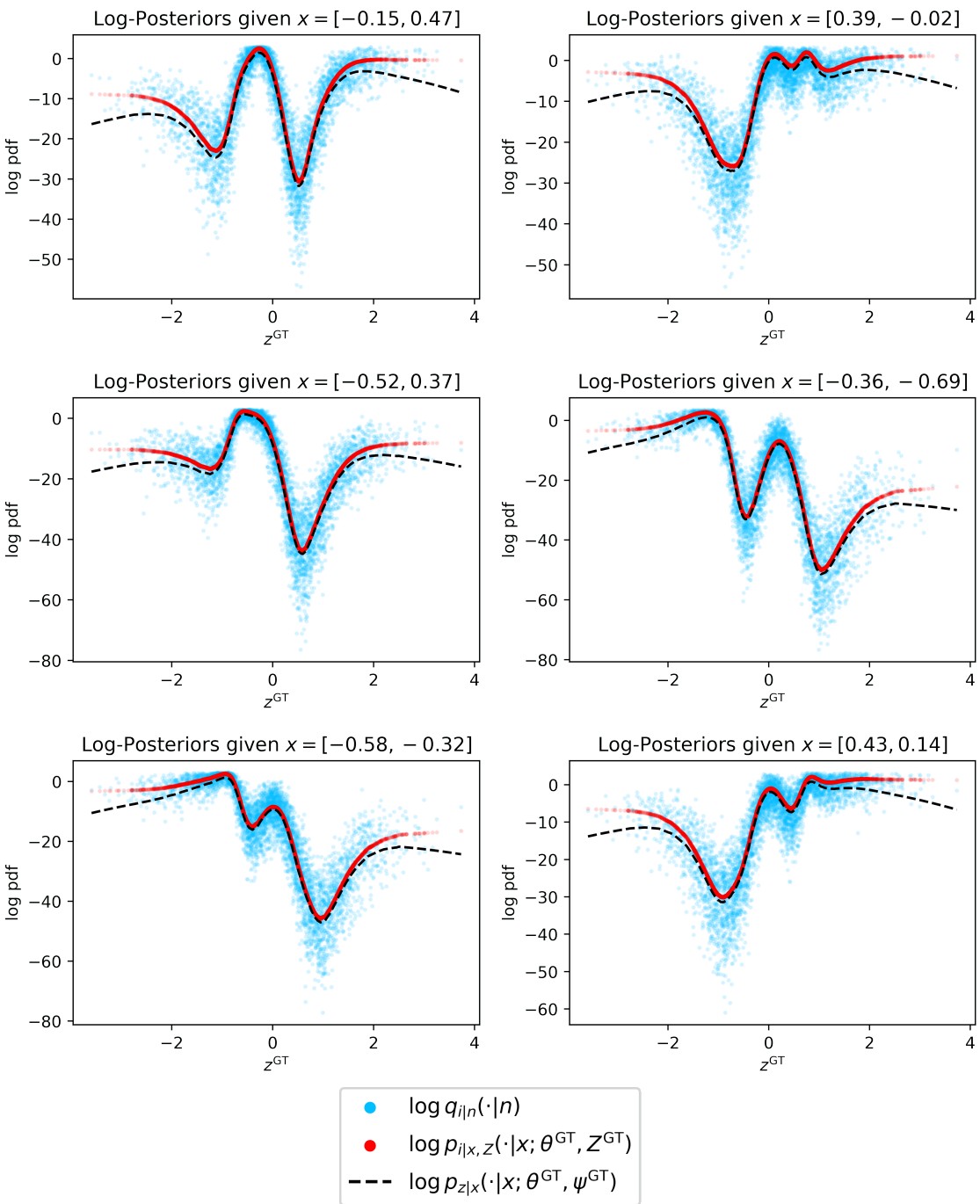

Figure 9: **MAPA captures trend of ground-truth posterior on "Figure-8" Example.** The panels compare the log-posteriors (of different $x$'s) vs. $z^{\mathrm{GT}}$. The black, red and blue represent the true posterior of the original model, the true posterior of the empiricalized model, and the MAPA, respectively. Details in Appendix D.3.

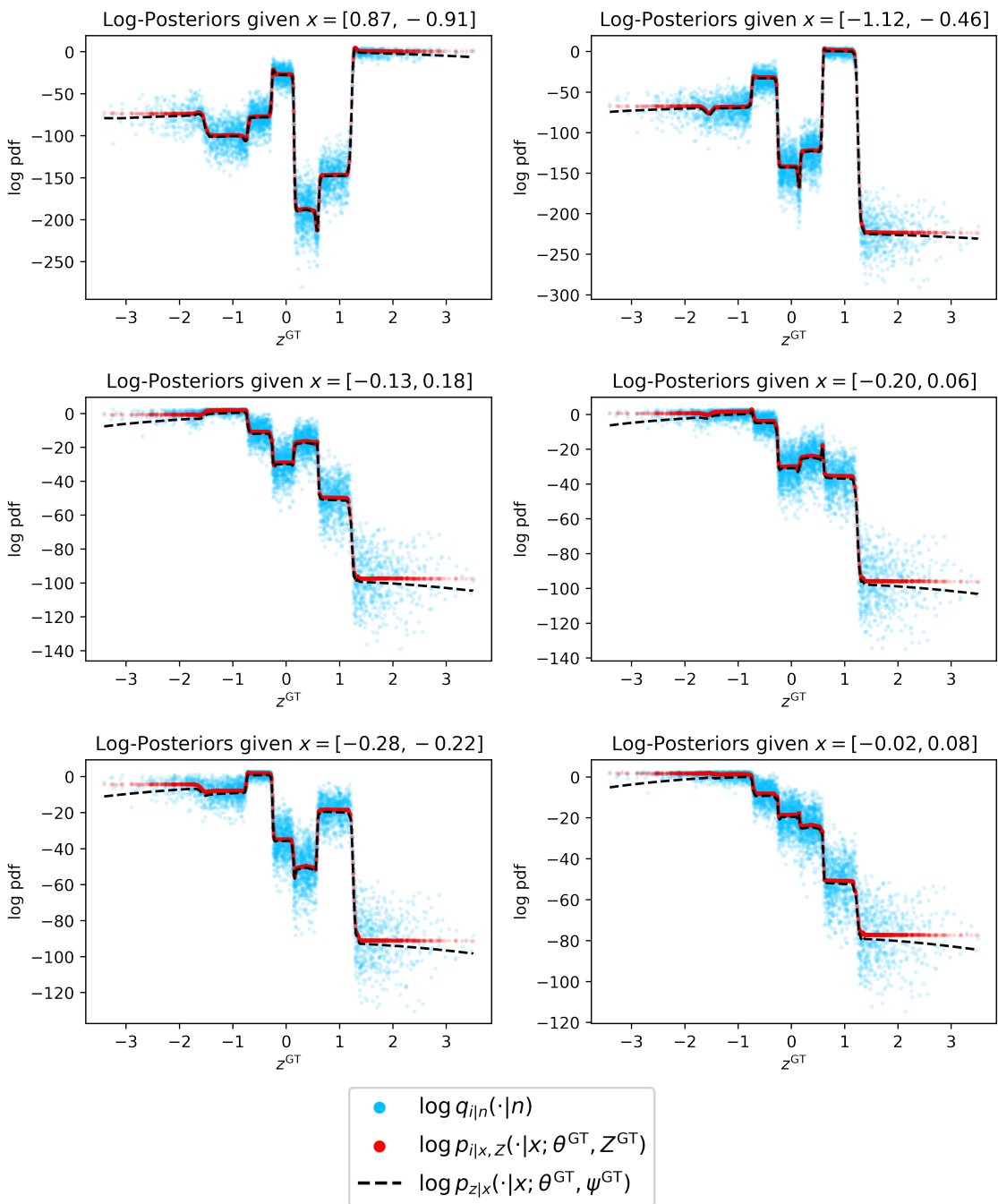

Figure 10: **MAPA captures trend of ground-truth posterior on "Spiral-Dots" Example.** The panels compare the log-posteriors (of different $x$'s) vs. $z^{\text{GT}}$. The black, red and blue represent the true posterior of the original model, the true posterior of the empiricalized model, and the MAPA, respectively. Details in Appendix D.3.

### D.4. MAPA is robust to model non-identifiability

In Appendix D.3, we showed that MAPA captures the trend of the ground-truth posterior. But if several models explain the observed data equally well, can MAPA capture the posterior trends in all? Given two different decoders $f_{\theta^{\mathrm{GT}}}(\cdot) \neq f_{\hat{\theta}}(\cdot)$ that induce the same true data distribution $p_x(\cdot; \theta^{\mathrm{GT}}) = p_x(\cdot; \hat{\theta})$, MAPA captures the trend in both equally well. We designed the following experiment to show this:

1. We selected two data-sets—the "Absolute-Value" and "Circle" examples—for which a VAE can estimate the data distribution accurately, but for which the inductive bias of the mean-field Gaussian variational family prevents it from recovering the ground-truth $f_\theta(\cdot)$ (Yacoby et al., 2020b).

2. For these data-sets, we call the ground-truth data-generating model "Variant 1" and the equally-good, learned model "Variant 2."

3. We confirm that the two variants indeed have different decoders $f_\theta(\cdot)$ by visualizing them in the top-rows of Figs. 11 and 12.

4. Now, we compare how well MAPA approximates the posteriors of each variant (as done in Appendix D.3) on the same selection of points (each $x$ gets its own row). Note: since we get Variant 2 by training a VAE, we cannot plot its log posterior relative to $z^{\mathrm{GT}}$. We instead use means of the mean-field Gaussian posterior approximations.

Figs. 11 and 12 show the result of this experiment: MAPA is robust to model non-identifiability—it is computed *once* per data-set, *but yields equally-good approximations on both variants.*

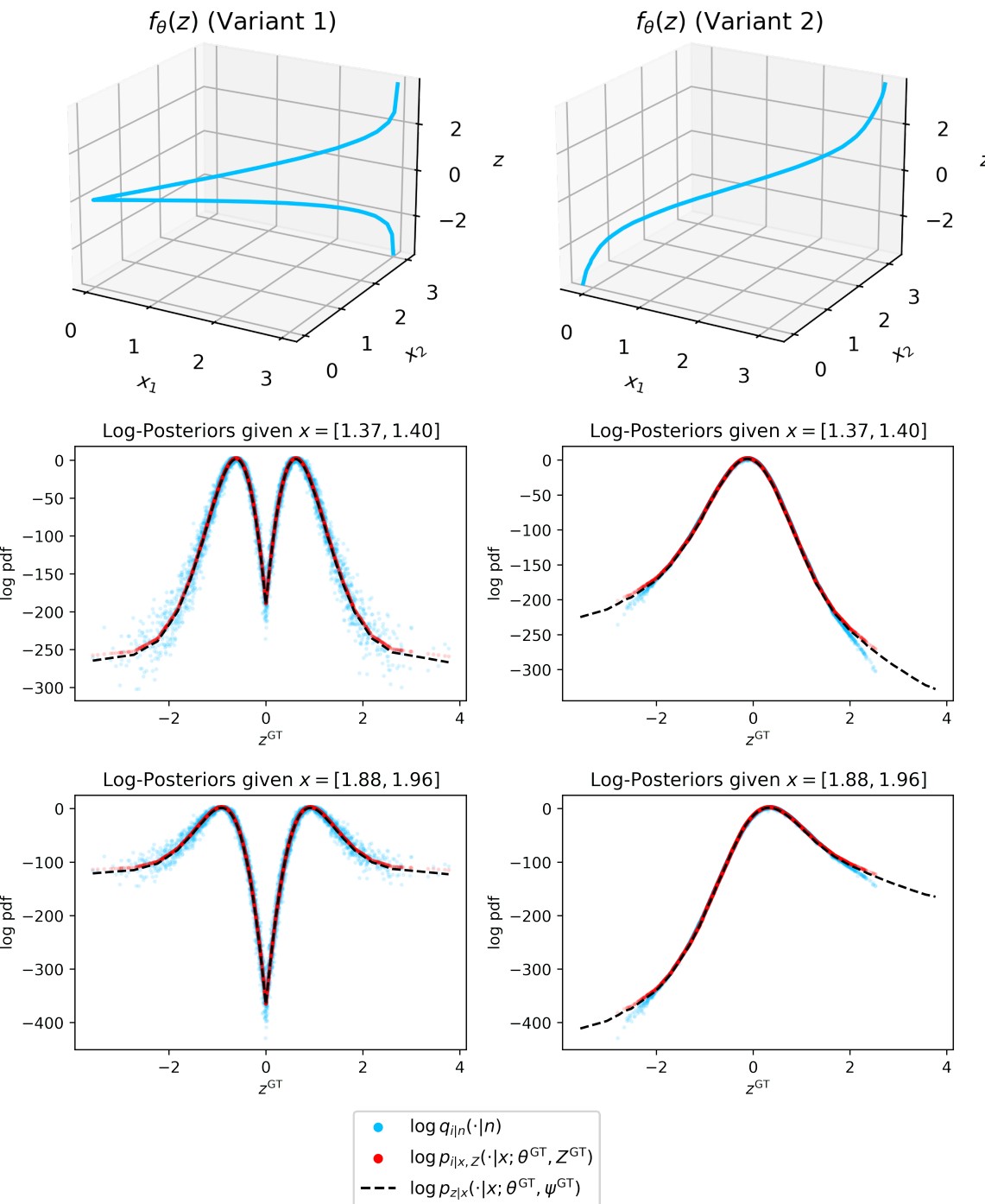

Figure 11: **MAPA is robust to model non-identifiability on "Absolute-Value" Example.** Top-row: Two generative models with different $f_\theta(\cdot)$ that yield the same $p_x(\cdot;\theta)$. Under them, the left and right columns compare the MAPA approximation to the ground-truth posteriors for each of the two models, respectively, on the same $x$'s. MAPA captures the trend in both equally well.

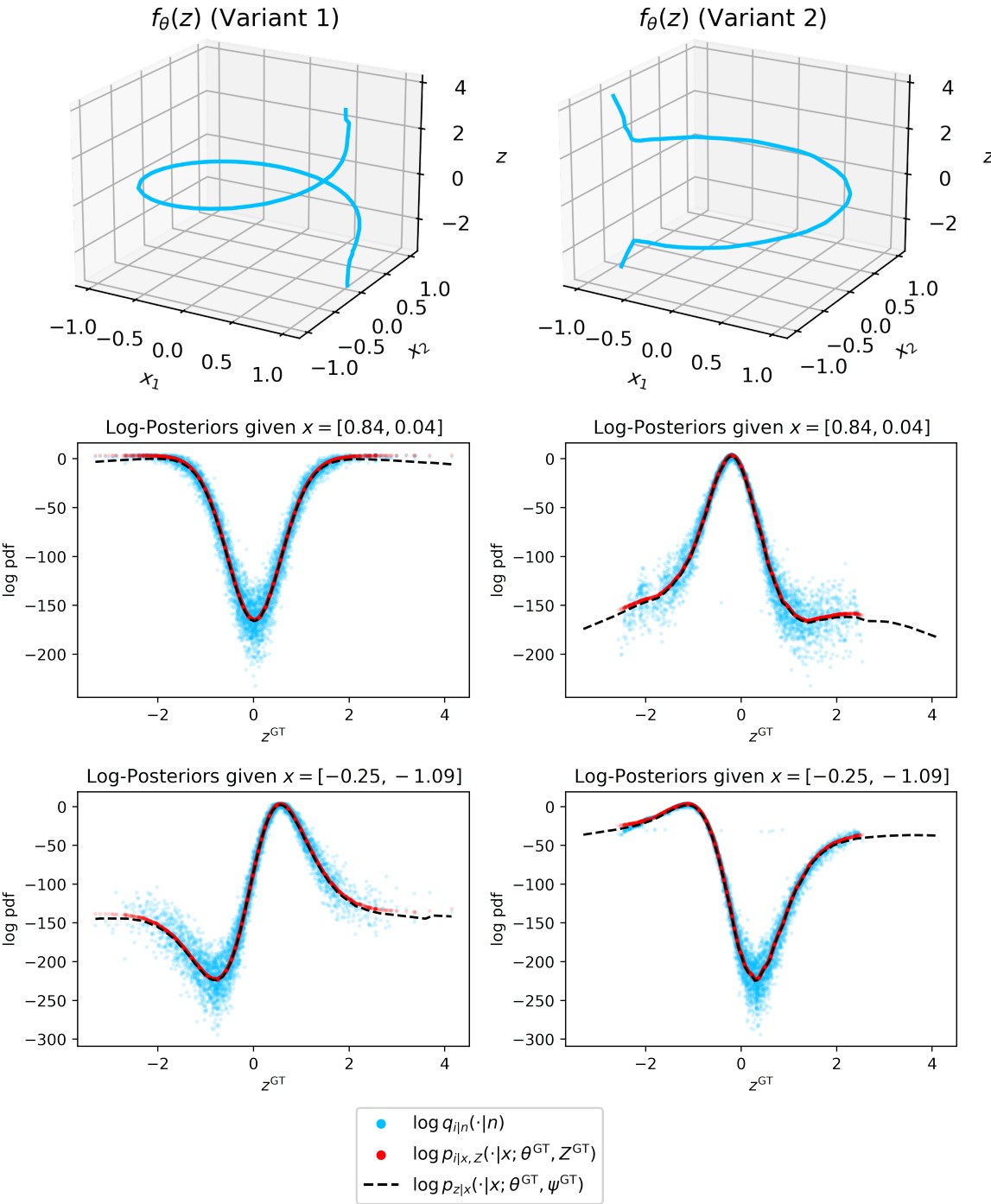

Figure 12: **MAPA is robust to model non-identifiability on "Circle" Example.**
Top-row: Two generative models with different $f_\theta(\cdot)$ that yield the same $p_x(\cdot; \theta)$.
Under them, the left and right columns compare the MAPA approximation to
the ground-truth posteriors for each of the two models, respectively, on the same
$x$'s. MAPA captures the trend in both equally well.

