# OpenReview forum: "Towards Model-Agnostic Posterior Approximation for Fast and Accurate Variational Autoencoders"
_approximateinference.org/AABI/2024/Symposium — AABI 2024_

### Official Review · Reviewer_KDQm · 2024-04-23
**New formulation VAE model and training**

**Rating:** 8
**Confidence:** 5

**Review:**

Summary:
This paper proposes a novel formulation of the VAE objective, given parametric generative process as in standard VAE, but with novel components: parametric (amortised) prior distribution, and non-parametric formulation of posterior distribution. The resulting variational objective is again a lower bound on the log-likelihood. Through empirical evaluations on synthetic dataset, the proposed model lead to improvement in both accuracy and efficiency comparing with VAE and IWAE.

Pros:
- The idea presented in this paper is truly novel, in replacing the amortised posterior with its non-parametric counterpart, the authors show that the resulting model lead to more efficient and model-agnostic density estimation and posterior inference.
- The posteriors, despite being non-parametric, accurately captures the true posterior. I hope the authors could continue working on clarifying the theoretical aspects of such posterior approximation accuracy/error.

Questions:
- In Page 2, the authors claims that "the empiricalized model is similar in spirit to bootstrapping", could the authors elaborate on the connections?

Minor points:
- From a high level perspective, the proposed MAPA framework is similar to the Recognition Parametrised Model (RPM; Walker et al. 2023) in their semi-parametric formulation of variational objectives in latent variable models. The authors could consider discussing the relationship of the MAPA model and the RPM model.

Reference:
Walker, W.I., Soulat, H., Yu, C. and Sahani, M., 2023, April. Unsupervised representation learning with recognition-parametrised probabilistic models. In International Conference on Artificial Intelligence and Statistics (pp. 4209-4230). PMLR.

---

### Official Review · Reviewer_Fxih · 2024-04-23
**Training VAE generative models using fixed posterior approximations**

**Rating:** 5
**Confidence:** 3

**Review:**

The work is original and written somewhat clear. They also show that MAPA (their method) is faster to converge than other VAE training methods. However, there seems to be some points that I think need further work.

1) The plots in Figure 1 use 10 random draws of the dataset and only plot the mean performance for the 10 draws. It then claims that MAPA outperforms baselines which is not clearly shown in the figure. IWAE seems to do quite well in all experiments. Furthermore, without error bars it is hard to tell if any of the very slight differences between the models is significant.

2) MAPA assumes an RBF kernel (for gaussian observation noise) to find the model-agnostic posterior. For this method to work in high dimensions such as images, it requires extremely dense data because the images need to be close in pixel space (and k of them need to be). Otherwise, all the kernels would be nearly zero. Increasing the variance would not work as this would give a poor "true" generative model of images. This seems like poor scaling from the simple problems shown in the paper.

3) The paper does not mention how one would get posteriors on z from the trained VAE. g(x_i) gives deterministic z_i and not a distribution. MAPA gives only a posterior distribution over i.

---

### Official Review · Reviewer_cYY2 · 2024-04-24
**Novel approximate inference for VAEs**

**Rating:** 7
**Confidence:** 4

**Review:**

This work proposes an alternative VAE inference algorithm that trains the generative and inference models independently.

To that end, a method to approximate the posterior of the true model a priori is presented, by leveraging an "empiricalized" VAE model.
- The authors show that a deterministic, model-agnostic posterior approximation (MAPA) of the true model’s posterior can be computed, based on the insight that an empiricalized model enables estimation of the probability of a latent code’s index $i_n$ independently of its location in latent space $z_n$.
- Namely, that the posteriors of nearby observations should score their corresponding latent code with high probability, a behavior that shall hold across multiple decoders.
- Hence, they propose MAPA as in Equation (8), which is a form of Kernel Density Estimator that uses distances in data-space to approximate a posterior distribution over the latent (lower-dimensional) space.

Based on MAPA, then a variational approximate inference method is presented, where an "approximate" MAPA posterior is used for computational purposes (i.e., only top $k$ MAPA indexes are used in the stochastic variational lower-bound computation).
- Minor question: right above Equation (9), is the sentence "after setting the probability of its $k$ largest elements to 0" correct? Isn't Equation 9 implying the contrary, i.e., "after setting the probability of its non-largest elements to 0"?

The authors present results on low-dimensional synthetic datasets, where they show that MAPA outperforms baselines on density estimation across sampling budgets $S$. Notably, they also showcase the importance of using the proposed MAPA posterior and how accurate it is in recovering the posterior of the ground-truth model. Additionally, the authors showcase the computational benefits of the proposed inference.

Overall, the work is well written, and presents a novel and interesting alternative towards more accurate and faster inference in VAE models. However, presented results are limited to low-dimensional synthetic examples. Interestingly, the authors also present a roadmap for scaling the MAPA-based inference method to high-dimensional data, which may foster novel insights and discussions within the workshop.

---

### Official Review · Reviewer_SSDf · 2024-04-25

**Rating:** 5
**Confidence:** 3

**Review:**

In this paper the authors propose a new method for fitting VAEs.  The key idea is to replace the continuous prior on the latent space with a purely atomic "empicalized" prior.  This converts the problem of finding the posterior distribution to one of finding which prior points most likely generated the observed data points.  The authors then use a trick where the number of atoms in the prior matches the number of observations, and assigns each point in the prior to an observation.  This allows the authors to make an ansatz that observations that are similar in data space likely arose from points that are nearby in the latent space.  This allows them to only need to fully consider a small number of potential points in the prior that could have generated the focal observation (and the rest of the prior atoms are dealt with via importance sampling). The authors use this approach on some toy datasets as a proof-of-concept and show that their method is competitive in terms of accuracy and faster than VAEs and IWAE.

I am a bit torn about this submission.  I find the overall idea very intriguing and the connection to standard auto-encoders is really nice.  The empirical performance is also compelling.  Yet, I find the connection to Bayesian statistics tenuous at best.  There are a number of approximations that, while intuitive, seem generally unjustified to me and it's not obvious that this approach is optimizing an objective that results in the learned approximate posterior being an optimal approximation to the true posterior in any concrete sense (e.g., like how VI can be shown to be minimizing the reverse KL between the approximate and true posteriors).  In some sense, this point is a bit moot because the method seems to work in practice (and so who cares if it's learning a true posterior or not), but I do think that the text sort of oversells the connection to Bayesian inference.  I have a number of other comments that I list below in the hopes that they're useful to the authors:

- Does the prior on $Z$ appear anywhere in the MAPA objective?  I don't believe it does which makes me skeptical about connections to general statistical inference.  For instance in the notation of the "Model" section, we may want to consider $\psi$ fixed and only maximize the LML with respect to the likelihood model (i.e., optimize over $\theta$).  Yet, that optimum will (in the true generative model) depend on what $\psi$ is, but if I understand correctly, this does not appear anywhere in the MAPA objective (equation 10).
- The stochastic IWAE bound also depends on $\phi$ (which is not reflected in the notation) and is optimized over $\psi$, $\theta$, and $\phi$.
- In Equation (3) why is there a prior on $z_m$ in the empircalized model?  Throughout the remainder of the section the $z$'s are just atoms (and it doesn't matter what distribution they were drawn from).  At no point (as far as I can tell) is the population from which they're drawn integrated over.  E.g., $g_\psi(\cdot)$ returns a particular $z_m$, not a distribution over the latent space.  That is, in some sense in the empiricalized model the $z$'s are hyperparameters as opposed to latent variables.
- The discussion around equation (8) is a bit confusing notationally.  I believe $k(x_n| x_i)$ is more like an "affinity" than a distance -- when it is large, it is assumed that the probability that $x_n$ came from $z_i$ is large (because $x_n$ and $x_i$ are _close_).  That is, $k$ is large when the points are close, and $k$ is small when the points are far. Calling $k$ a distance is confusing because of this.
- The authors say that the posterior approximation is deterministic, but I'm not sure what they mean by this.  During training, they use importance sampling to integrate over their approximate posterior (non-deterministic), and in general the posterior over the atoms puts mass on more than one point, so any given data point could have been generated by multiple atoms.
- I was also confused by Equation (14).  A log-likelihood term is referred to as the "autoencoder loss", but it's actually the negative loss, I believe (i.e., autoencoders _maximize_ the log-likelihood, so they would minimize the negative of this as their loss).  Furthermore, because of this, it's not clear to me what Equation 14 is showing.  Maximizing the MAPA objective could be achieved by improving the autoencoder loss or _maximizing_ the "gap" and it is not obvious to me that the "gap" is bounded. Contrast this to the VI loss where the ELBO is the evidence _minus_ a strictly non-negative term.
- I am wondering how this approach performs for high dimensional observed data.  In high dimensions it's easier for points to be essentially equidistant to all other points, making the kernel approach to estimating which points likely contribute to the posterior less useful.
- Perhaps I missed it, but how many observations were used in the simulations?
- Comparing to IWAE and VAE on the forward KL in Supp, D1 seems a bit unfair as IWAE and VAE are optimizing the reverse KL, so they're not expected to be optimal in terms of the forward KL.
- Presumably the results in Figures 4 and 5 depend on the dataset size.  If the dataset is huge, then you would need many more samples before starting to see repeatedly sampled indices across the batch.
- A minor point, but is sampling done with or without replacement in the importance sampling bit?  Presumably if you are sampling enough points to be getting repeated draws across the batch, then you are also sampling enough points were sampling without replacement would result in lower variance than sampling with replacement.
- I was not sure what was getting plotted in Figures 6--10.  The blue points are labeled as $q_{i|n}$ but that is a distribution over indices, not a distribution on the space of $z$s.
- An alternative view of / motivation for this framework could eschew the connection to statistical inference altogether, and instead frame the intuition in terms of the autoencoder.  In particular, one can view this approach as somehow regularizing the encodings of points that are nearby in the observed space to result in latent representations that result in similar mappings back to the observed space (i.e., if x_i and x_j are close, then Decoder(Encoder(x_j)) should err on the side of being between x_i and x_j in order to minimize the MAPA loss).  To me this motivation is more principled in that it's a form of regularization, whereas the motivation based on Bayesian inference that is presented in the paper is less clear to me for the reasons listed above.

Typos:
- "Second, it provides a convenient mapping to and from latent space" --> "Second, it provides a convenient mapping to and from the latent space"
- "MAPA is also bears similarity to Approximate Bayesian Computation" --> "MAPA also bears similarity to Approximate Bayesian Computation"

---

### Official Review · Reviewer_6JtQ · 2024-04-25
**An interesting, novel approach and a well-written paper with preliminary results and open questions about effectiveness**

**Rating:** 7
**Confidence:** 4

**Review:**

The paper proposes a VAE inference algorithm that involves training the generative and inference models independently. The strategy involves the introduction of an empiricalized generative model, and the construction of an approximation of the posterior over the indices of the samples. This is a categorical distribution and requires the introduction of a similarity function (kernel). Based on the constructed model, the paper derives a stochastic lower bound to the log marginal likelihood of the empiricalized model. Inference then involves maximizing this bound and learning a parametric prior distribution.

Strengths:

(1)	The paper presents a interesting approach that is novel (to the best of my knowledge).

(2)	The paper is well-written; there is a clear motivation for the proposed approach, and the development of the technique is easy to follow.

(3)	The preliminary proof-of-concept experiments illustrate that the proposed method has potential.

Weaknesses:

The paper is submitted as a preliminary work, so some of the weaknesses specified below stem from that.

(1)	There are no experiments on high-dimensional real-world data, which is really the domain of most interest for VAEs. The paper suggests some strategies for addressing the scaling issue, but most of these involve approximations of some form, and it is unclear whether or not the introduction of such approximations will eliminate the benefits that motivated the development of the approach.

(2)	The specified method requires the specification of a similarity function (or kernel). The paper derives this for Gaussian observation noise and Bernoulli likelihoods. In general, the observation noise will not be known (and will not follow a nice parametric form). In such circumstances, the choice of a suitable kernel may be challenging.

(3)	Given the success of diffusion models, even in high dimensions, with an acceptable computational overhead, it would be useful to see a discussion of why the variational auto-encoder approach, and particularly the approach presented here, would be considered advantageous. The proposed technique involves training a neural network and (potentially) normalizing flow – if the denoising diffusion techniques are outperforming the VAE/normalizing flow methods, then why concentrate on them for the empiricalized inference strategy.

---

### Meta-Review · Area_Chair_S3GZ · 2024-05-24

**Recommendation:** Accept (Poster)
**Confidence:** 3

**Metareview:**

This paper proposes a novel method for learning VAE models. The reviews were somewhat polarised in the scores, with a mix of clear accepts and marginal rejects. The rejects themselves were somewhat conflicted. Overall, however, there is consensus that the paper is interesting, original, and largely clearly written. A common criticism is that the experiments are too preliminary. However, I believe this paper is worth sharing with the community as it will encourage interesting discussions and these ideas could be iterated on in future work. Furthermore, I think the bar for experimental validation required for the extended abstract track has been reached. Thus, I recommend that the paper be accepted.

---

### Decision · Program_Chairs · 2024-05-27

Accept